# MoFA: Dual-Task Motion Factorization for Human Motion Synthesis with Imperfect and Limited Data

## Abstract

Human motion synthesis has recently benefited from diffusion models, achieving unprecedented realism and diversity. Yet precise and controllable generation remains challenging: text, audio, and 2D cues are often ambiguous, while existing trajectory-keyframe approaches suffer from limited generalization, naive feature fusion, and poor robustness to unpaired control signals. We identify this bottleneck as the entanglement between keyframe and trajectory signals, which are inherently coupled in training but frequently mismatched at inference. To address this, we propose **MoFA**, a diffusion-based **Mo**tion **Fa**ctorization framework that decomposes synthesis into two complementary sub-tasks: (i) *Local Motion Completion*, focusing on keyframe dynamics, and (ii) *Trajectory Adaptation*, ensuring global spatial consistency. MoFA integrates the *Local Motion Refinement Stack (LMRS)* and the *Trajectory-Aware Motion Integration (TAMI)* to jointly refine local poses and adapt them to trajectories. In addition, we introduce a *Quality-Aware Dual Training (QADT)* strategy that leverages imperfect or low-quality data as auxiliary supervision, substantially expanding the effective training set and improving generalization. Extensive experiments demonstrate that MoFA achieves more stable, controllable, and robust motion synthesis than advanced baselines.

## 1 Introduction

Human motion synthesis is a fundamental problem in computer animation and virtual interaction. Early approaches based on VAEs (Petrovich et al., 2022; Wang, 2023; Zhang et al., 2023; Lu et al., 2023; Ma et al., 2024; Tevet et al., 2022a; Guo et al., 2024) and GANs (Xu et al., 2023; Zhou et al., 2024; Li et al., 2024) achieved important progress in realism and controllability. More recently, diffusion models (Ho et al., 2020; Song et al., 2020; Ho et al., 2022) have emerged as the state-of-the-art, significantly advancing the naturalness and diversity of synthesized motions. Within this paradigm, text (Athanasiou et al., 2024; Chi et al., 2024; Dabral et al., 2023; Xie et al., 2024; Tevet et al., 2022b; Chen et al., 2024b; Liao et al., 2025; Lee et al., 2025; Zhang et al., 2025), audio (Qiu et al., 2025; Xu et al., 2025; Wang et al., 2025a; Li et al., 2025), and 2D visual cues (Zhong et al., 2025; Wang et al., 2025b) have been explored as control signals, enabling cross-modal motion generation.

Despite these advances, precise and controllable motion synthesis remains challenging. Text prompts, though intuitive, are often ambiguous and imprecise, leading to inconsistent outputs. Audio and 2D inputs also suffer from modality gaps, limiting their reliability in fine-grained control. To overcome these limitations, recent works incorporate explicit signals such as trajectories and keyframes(Xi et al., 2025): trajectories provide global spatial guidance, while keyframes impose local pose constraints. However, existing methods face three critical issues: (i) limited generalization due to low-diversity datasets, (ii) naive fusion of trajectory and keyframe features, often producing conflicts or temporal inconsistencies, and (iii) poor robustness to unpaired control signals, resulting in unstable or unrealistic outputs.

We argue that these challenges stem from a deeper issue: the entanglement between keyframe and trajectory signals. Current models assume independence between them, yet in practice they are inherently coupled-during the training process, the keyframes' postures are actually aligned to the

positions in trajectories. This mismatch between training assumptions and data structure leads to overfitting and poor generalization: because keyframes and trajectories are paired during training, unpaired control signals at inference time may result in degraded output quality.

To address this, we propose **MoFA**, a diffusion-based **Mo**tion **Fa**ctorization framework. Our key idea is to decompose motion synthesis into two sub-tasks: (1) *Local Motion Completion*, focusing on local motion synthesis without trajectory, and (2) *Trajectory Adaptation*, adding global information and ensuring global spatial consistency. We realize this via two complementary modules: the *Local Motion Refinement Stack (LMRS)* and the *Trajectory-Aware Motion Integration (TAMI)*. Besides, to enhance robustness and generalization, we introduce a *Quality-Aware Dual Training (QADT)* strategy, which leverages imperfect or low-quality data as auxiliary supervision to enrich the learned feature domain. This allows MoFA to better handle challenging scenarios with improved stability and quality.

Our contributions are threefold:

- We propose a **Motion Factorization** method that decomposes motion synthesis into two complementary sub-tasks: local motion completion and trajectory adaptation. This factorization explicitly disentangles local keyframe alignment from global trajectory consistency, allowing the model to first learn approximate local motions and then adapt them to trajectories for complete global motion generation. This design enhances stability and robustness, particularly when handling unpaired control signals at inference time.

- We design the **MoFA framework**, a novel dual-task diffusion model built on a transformer architecture. It integrates two complementary modules, LMRS for local refinement, and TAMI for global adaptation, which is dedicated to a specific sub-task. The dual-branch design enables the model to naturally accommodate different training objectives, thereby improving performance on global motion completion.

- We design a **Quality-Aware Dual Training (QADT)** strategy that effectively leverages both high-quality and low-quality motion datasets. By sharing network modules, high-quality data jointly supervises local and global motion generation, while low-quality data with unreliable trajectories is utilized for local motion learning through a trajectory-agnostic embedding. This dual-task training scheme substantially expands the usable training set and improves model generalization.

## 2 RELATED WORK

Early approaches to human motion synthesis relied on probabilistic and deep generative models such as normalizing flows (Henter et al., 2020), conditional VAEs (Guo et al., 2020), and latent alignment frameworks (Petrovich et al., 2022). The release of large-scale benchmarks such as HumanML3D (Guo et al., 2022) and Combat (Yihao Liao, 2024) further standardized evaluation and enabled rapid progress in text-to-motion (T2M). More recently, diffusion models have become the dominant paradigm, with works such as (Zhang et al., 2024; Tevet et al., 2022b; Huang et al., 2024) demonstrating strong performance in realism and diversity, while diffusion priors (Shafir et al., 2023) facilitated long-sequence and interactive generation. Building on this foundation, subsequent studies have emphasized long-term temporal consistency, robustness, and disentangled control. For instance, recurrent and hierarchical extensions improve sequential modeling (Mohamed et al., 2024; Sun et al., 2024), while alternative formulations such as masked or autoregressive diffusion have been explored to enhance controllability (Meng et al., 2024). Efficiency and stability have also been addressed through lightweight or robust pipelines (Huang et al., 2024; Cohan et al., 2024).

Beyond text-only conditioning, multimodal signals have been increasingly incorporated to enable finer-grained synthesis. Representative works integrate trajectories, keyframes, or scene contexts into the generative process (Wan et al., 2024; Xi et al., 2025; Geng et al., 2024; Zhao et al., 2025; Zhang et al., 2025), significantly broadening the applicability of motion synthesis. In particular, explicit control via trajectories and keyframes has emerged as a promising strategy to balance global spatial guidance with localized pose specification (Xi et al., 2025; Zhao et al., 2025; Bae et al., 2025). However, existing approaches typically treat keyframes and trajectories as independent modalities and attempt to fuse them at the feature level. This assumption overlooks their inherent dependency: in the training set, trajectories already influence the specified keyframe poses, creating implicit cor-

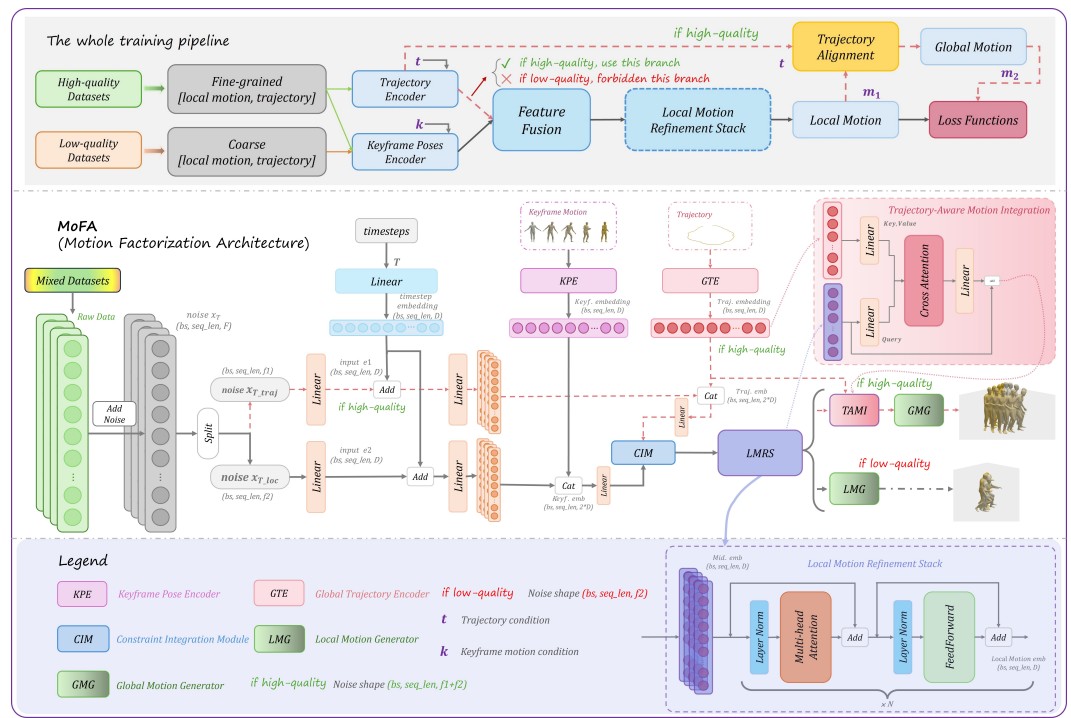

Figure 1: Overall architecture of the proposed **MoFA** framework. The top row illustrates the full training pipeline, where the Quality-Aware Dual Training (QADT) strategy leverages both high-quality and low-quality datasets. The medium row illustrates that keyframe and trajectory control signals are separately encoded and then refined and fused through the Constraint Integration Module (CIM) and the Local Motion Refinement Stack (LMRS) to obtain local motion features. Depending on the training setting, the model either (i) generates local motion sequences via the Local Motion Generator (LMG), or (ii) performs trajectory alignment through the Trajectory-Aware Motion Integration (TAMI) module, followed by the Global Motion Generator (GMG) to produce complete global motions. The bottom row is the legend.

relations between the two signals. As a result, naive decoupling and simple fusion strategies can introduce overfitting, reduce temporal consistency, and hinder the model's ability to generate coherent and complete motion sequences from unpaired control signals. These limitations highlight the need for new frameworks that explicitly consider the entanglement between trajectory and keyframe guidance, enabling more robust, adaptive, and controllable human motion synthesis.

# 3 METHOD

## 3.1 PRELIMINARIES OF THE DIFFUSION

The Diffusion Model (DM) learns a diffusion process that generates a probability distribution for a given dataset. In the case of visual content generation tasks, a neural network of DM is trained to reverse the process of adding noise to real data so new data can be progressively generated starting from random noise. For a data sample $\mathbf{x} \sim p_{\text{data}}$ from a specific data distribution $p_{\text{data}}$, the forward diffusion process is defined as a fixed Markov Chain that gradually adds Gaussian noise to the data following:

$$q\left(\mathbf{x_t} \mid \mathbf{x}_{t-1}\right) = \mathcal{N}\left(\mathbf{x}_t; \sqrt{1-\beta_t}\mathbf{x}_{t-1}, \beta_t \mathbf{I}\right) \tag{1}$$

for $t = 1, \cdots T$, where $T$ is the number of perturbing steps and $x_t$ represents noisy data after adding $t$ steps of noise on the read data $x_0$. This process is controlled by a sequence schedule $\beta_t$ which is parameterized by the noising step $t$. Following the closure of normal distribution, $x_t$ can be directly computed with $x_0$ by reforming the above diffusion process as follows:

$$q\left(\mathbf{x_t} \mid \mathbf{x}_0\right) = \mathcal{N}\left(\mathbf{x}_t; \sqrt{\bar{\alpha}_t}\mathbf{x}_0, \left(1-\bar{\alpha}_t\right)\mathbf{I}\right) \tag{2}$$

where $\bar{\alpha}_t = \prod_{i=1}^{t} \alpha_i$ and $\alpha_t = 1 - \beta_t$. Following DDPM, a denoising function $\epsilon_\theta$ parameterized with $\theta$, commonly implemented with a neural network, is trained by minimizing the mean square error loss as follows:

$$\mathbb{E}_{\epsilon \sim \mathcal{N}(\mathbf{0},\mathbf{I}),\mathbf{x}_t,\mathbf{c},t}\left[\left\|\epsilon - \epsilon_\theta\left(\mathbf{x}_t; \mathbf{c}, t\right)\right\|_2^2\right] \tag{3}$$

where $\mathbf{c}$ is an optional condition and $\mathbf{x}_t$ is a perturbed version of real data $\mathbf{x}_0 \sim p_{\text{data}}$ by adding $t$-step noises. In this way, $\epsilon_\theta$ can be trained till converge by sampling $\mathbf{x}_0$ from real data distribution and a time step $t$, with an optional condition $\mathbf{c}$.

## 3.2 MOTION FACTORIZATION METHOD

We formulate motion generation as a structured probabilistic modeling problem. Let $t$ denote the global trajectory control, $k$ the keyframe control, $m_1$ the complete local motion without $t$, and $m_2$ the complete global motion.

Conventional approaches directly learn a mapping $f(t, k) \to m_2$, which entangles trajectory and keyframe constraints into a single multimodal distribution. Instead, we introduce an intermediate local motion representation and assume that keyframes influence the global motion only through this intermediate variable. Formally, we assume the conditional independence

$$m_2 \perp k \mid (t, m_1) \tag{4}$$

which leads to the factorization

$$P(m_2 \mid t, k) = \int P(m_2 \mid t, m_1) \, P(m_1 \mid t, k) \, dm_1 \tag{5}$$

This decomposition naturally induces a two-stage generation process:

$$m_1 \sim P(m_1 \mid t, k) \tag{6}$$
$$m_2 \sim P(m_2 \mid t, m_1) \tag{7}$$

The first stage, *Local Motion Completion*, predicts a distribution of plausible local motions conditioned on $(t, k)$. The second stage, *Trajectory Adaptation*, generates a globally consistent motion sequence conditioned on $(t, m_1)$.

Given training samples $(t, k, m_1, m_2)$, the joint log-likelihood can be decomposed as

$$\log P(m_1, m_2 \mid t, k) = \log P(m_1 \mid t, k) + \log P(m_2 \mid t, m_1) \tag{8}$$

Thus the optimization objective becomes

$$\mathcal{L} = \mathbb{E}_{(t,k,m_1,m_2)} \big[ -\log P_\theta(m_1 \mid t, k) - \log P_\phi(m_2 \mid t, m_1) \big] \tag{9}$$

where $\theta$ and $\phi$ parameterize the local-motion and global-motion generators, respectively.

This two-stage motion factorization method offers three benefits: (i) **Reduced complexity**: the multimodal distribution $P(m_2 | t, k)$ is simplified into two conditional distributions; (ii) **Interpretability**: the latent local motion $m_1$ serves as an explicit bridge between keyframes and full-body motion; (iii) **Robustness**: perturbations in keyframes mainly affect $m_1$, while the second stage regularizes global structure through the trajectory $t$. The two stages are mutually supportive to each other.

## 3.3 QUALITY-AWARE DUAL TRAINING STRATEGY

High-quality motion datasets are relatively scarce. For instance, HumanML3D (Guo et al., 2022) provides reliable annotations but remains limited in scale. In contrast, many available datasets contain reasonably accurate *local motions* but lack coherent and realistic *global trajectories*, making them difficult to exploit under conventional training pipelines. Our two-stage motion factorization framework provides a natural solution for leveraging such imperfect data. To this end, we introduce the **Quality-Aware Dual Training (QADT)** strategy, which unifies high-quality and low-quality data in a principled manner.

**Data Quality Partitioning.** We denote high-quality datasets as $D_h$ and low-quality datasets as $D_l$. To explicitly distinguish data quality during training, we assign quality marks $Q_h$ and $Q_l$ to each sample. The mark determines which generation tasks are activated and how the trajectory information is treated.

Figure 2: Qualitative comparison of our model against recent baselines under trajectory and keyframe pose joint guidance. Each block illustrates the provided trajectory (top-left), keyframe poses (bottom-left), the legends and the generated motion sequences. Our method produces motions that more faithfully follow the input trajectory and align with keyframe poses, while maintaining natural dynamics. Competing methods have unnatural body dynamics, as highlighted by the red dashed lines and emoticons.

**Local Motion Completion from Mixed Quality Data.** In the first stage, the model accepts both $Q_h$ and $Q_l$ samples. For $Q_h$, the model uses the trajectory $t$ together with keyframes $k$ to infer local motion $m_1$. For $Q_l$, however, the trajectory information is unreliable and thus discarded. Instead, we introduce a learnable embedding that serves as "trajectory", enriching the diversity of local motion distribution and strengthening the model's understanding of local dynamics. The output embeddings under both $Q_h$ and $Q_l$ conditions are passed through the Local Motion Generator to compute the local motion loss.

**Global Motion Completion with Reliable Trajectories.** For samples with $Q_h$, the reliable trajectory $t$ is further utilized in the second stage. Specifically, $t$ is integrated via the Trajectory-Aware Motion Integration (TAMI) module to condition the Global Motion Generator, which produces the final global motion $m_2$. In contrast, $Q_l$ samples do not enter this stage, since their global trajectories are not trustworthy.

**Diffusion Training Process.** During diffusion-based training, QADT realizes the dual-task structure by injecting different noise priors depending on the quality mark. For low-quality data ($Q_l$), we set the input as $X_t \in \mathbb{R}^{b \times l \times f_1}$, while for high-quality data ($Q_h$), the input is $X_t \in \mathbb{R}^{b \times l \times f_2}$, where $f_2 - f_1 = 6$, $b$ and $l$ represent the batchsize and sequence length of motion respectively, and the six additional channels encode trajectory information. In this way, high-quality samples guide the model to align local and global motion, whereas low-quality samples are utilized to regularize and diversify the learned local motion space.

### 3.4 MOFA FRAMEWORK DESIGN

The overall architecture of our proposed **MoFA** framework is shown in Figure 1. The design follows a diffusion-transformer (DiT) based generation paradigm and is tightly coupled with our Quality-Aware Dual Training (QADT) strategy, enabling the unified exploitation of both high-quality and low-quality datasets.

**Control Signal Encoding.** The input motion control signals consist of keyframe poses $k$ and global trajectories $t$. These are separately processed by two encoders: the *Keyframe Pose Encoder (KPE)* and the *Global Trajectory Encoder (GTE)*. The $t$ is replaced with a learnable trajectory em-

bedding $\tilde{t}$ to enrich the distribution for samples labeled $Q_l$. The decoupling of $k$ and $t$ ensures that the network captures localized pose constraints and global spatial guidance in a complementary manner:

$$z_k = \text{KPE}(k), \quad z_t = \text{GTE}(t). \tag{10}$$

**Constraint Integration and Local Motion Refinement.** The encoded features $(z_k, z_t)$ are fused through the *Constraint Integration Module (CIM)* and subsequently refined by the *Local Motion Refinement Stack (LMRS)*. LMRS, implemented as a transformer-based refinement module, generates coherent local motion representations:

$$m_1 = \text{LMRS}(CIM(z_k, z_t)) \tag{11}$$

This stage corresponds to the probabilistic factor $P(m_1|t, k)$ in our two-stage formulation, and is designed to remain robust under noisy or incomplete conditions.

**Dual Generation Paths.** MoFA explicitly supports two complementary generation paths under QADT:

- **Local Motion Learning:** For samples labeled $Q_l$ and $Q_h$, the local supervision is available. In this case, the *Local Motion Generator (LMG)* outputs localized motion sequences directly from $m_1$, $\hat{m}_1 = \text{LMG}(m_1)$.
- **Global Motion Learning:** For samples labeled $Q_h$, the reliable trajectory $t$ is further utilized. The *Trajectory-Aware Motion Integration (TAMI)* module aligns $m_1$ with $t$, and the *Global Motion Generator (GMG)* produces the full-body motion, $m_2 = \text{GMG}(\text{TAMI}(m_1, t))$.

### 3.4.1 LOSS FUNCTIONS

To jointly optimize the proposed framework, we design a composite loss function that integrates both reconstruction and regularization terms. The overall objective is defined as:

$$\mathcal{L} = \lambda_1 \cdot \mathcal{L}_{rot} + \lambda_2^* \cdot \mathcal{L}_{traj}^* + \lambda_3^* \cdot \mathcal{L}_{vel}^* + \lambda_4^* \cdot \mathcal{L}_{acc}^* + \lambda_5 \cdot \mathcal{L}_{joint}, \tag{12}$$

where each component serves a distinct role in constraining different aspects of motion generation: i) $\mathcal{L}_{rot}$: an $L_2$ reconstruction loss between the ground-truth and the generated local motion sequences. ii) $\mathcal{L}_{traj}$: an $L_2$ loss between the predicted and reference trajectories. This term is only activated when the quality mark is $Q_h$. We denote this conditional activation with the superscript $*$. iii) $\mathcal{L}_{vel}$ and $\mathcal{L}_{acc}$: smoothness regularizers that penalize discrepancies in first-order (velocity) and second-order (acceleration) derivatives, encouraging temporally coherent and physically plausible motion. iv) $\mathcal{L}_{joint}$: a joint-level reconstruction loss, where the joints are Cartesian coordinates (XYZ).

## 4 EXPERIMENTAL SETTING

**Datasets**: For global motion training, we employ two datasets: HumanML3D, containing 14,646 motion sequences, and CombatMotion, which serves as the base dataset with 14,883 sequences. In addition, for local motion training, we further incorporate AIST++ with 1,400 sequences and a subset of MotionX++ comprising more than 30,000 sequences to enrich the training set.

**Evaluation Metrics.** We evaluate our framework with a comprehensive set of metrics covering accuracy, diversity, and motion quality: (1) **MPJPE** (Mean Per Joint Position Error): the average Euclidean distance between generated and ground-truth joint values; (2) **Diversity**: the variance across multiple generated sequences under the same condition, reflecting generative variety; (3) **Trajectory Error**: the deviation between predicted and ground-truth global trajectories; (4) **K-MPJPE**: a variant of MPJPE computed at designated keyframes to assess fidelity at key poses; (5) **FID** (Fréchet Inception Distance): the distributional discrepancy between generated and real motion features. To further evaluate temporal coherence and physical plausibility, we additionally report: (6) **Joint Smoothness (JS)**: the smoothness of frame-to-frame joint transitions; (7) **Foot Sliding**: the degree of foot instability, measuring undesired ground-sliding artifacts; (8) **Motion Fluency**: the autocorrelation of generated sequences, quantifying the overall continuity and naturalness of the motion.

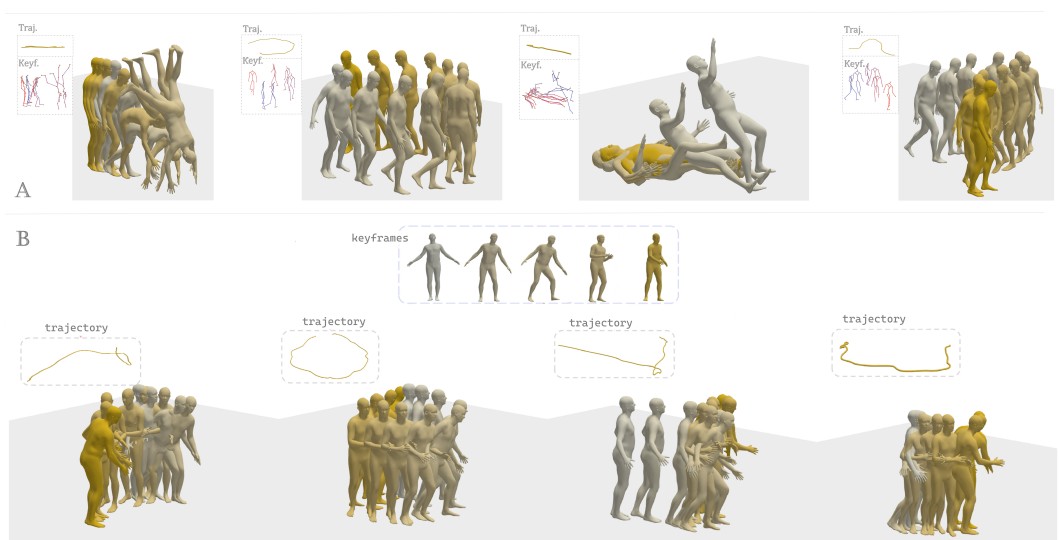

Figure 3: Visualized results of our MoFA under different guidance conditions. Figure A(top): Motion synthesis from various trajectory and keyframe inputs. Our model faithfully generates complete motion sequences, in normal scenaries like curved trajectories, in-place rotations or even in challenging cases such as complex backflips, and falling motions. Figure B(bottom): Motion synthesis with combined trajectory and keyframe pose guidance. The results demonstrate that our model effectively captures the relationship between trajectory orientation and coherent motion execution, producing plausible motions under different trajectory conditions given the same keyframe poses.

| Method | MPJPE ↓ | FID ↓ | Diversity → | Traj.err ↓ | K-MPJPE ↓ |
|---|---|---|---|---|---|
| CAMDM-5((Chen et al., 2024b)) | 5.354 ±.078 | 0.898 ±.034 | 4.464 ±.015 | 0.504 ±.129 | 5.187 ±.076 |
| MDM-5((Tevet et al., 2022b)) | 5.388 ±.083 | 0.751 ±.027 | 4.511 ±.011 | 0.390 ±.043 | 5.203 ±.082 |
| GMD-Unet-5((Guo et al., 2025)) | 11.918 ±.109 | 3.028 ±.039 | 4.434 ±.013 | 0.753 ±.079 | 11.805 ±.102 |
| GMD-Dit-5((Guo et al., 2025)) | 5.564 ±.099 | 0.729 ±.035 | 4.517 ±.017 | 0.212 ±.029 | 5.365 ±.095 |
| MotionCLR-5((Chen et al., 2024a)) | 11.003 ±.090 | 5.443 ±.129 | 3.792 ±.017 | 0.647 ±.076 | 10.937 ±.088 |
| MotionDiffuse-5((Dabral et al., 2023)) | 5.821 ±.066 | 0.566 ±.025 | 4.539 ±.016 | 0.465 ±.057 | 5.676 ±.065 |
| PriorMDM-5((Shafir et al., 2023)) | 7.917 ±.117 | 1.525 ±.058 | 4.370 ±.016 | 0.539 ±.114 | 7.813 ±.107 |
| StableMoFusion-5((Huang et al., 2024)) | 5.091 ±.093 | 0.301 ±.016 | 4.532 ±.018 | 0.418 ±.137 | 4.730 ±.087 |
| OmniControl-5((Xie et al., 2024)) | 5.469 ±.083 | 0.742 ±.026 | 4.482 ±.013 | 0.372 ±.069 | 5.324 ±.079 |
| PMG-5((Xi et al., 2025)) | 3.901 ±.076 | 0.324 ±.019 | 4.522 ±.019 | 0.085 ±.007 | 3.245 ±.069 |
| Ours-5 | **3.763** ±.083 | **0.293** ±.016 | **4.556** ±.012 | **0.078** ±.013 | **3.163** ±.072 |

Table 1: Quantitative comparison with advanced methods on multiple metrics, including MPJPE, FID, Diversity, Trajectory error, and K-MPJPE. Our method consistently outperforms all baselines, achieving the lowest errors (MPJPE, FID, Traj.err, K-MPJPE) while maintaining competitive diversity. Notably, our model achieves an MPJPE of 3.763 and FID of 0.293. The best results are highlighted in bold.

## 4.1 QUANTITATIVE AND QUALITATIVE RESULTS

### 4.1.1 COMPARED WITH OTHER MODELS

The Table1 reports a quantitative comparison with advanced methods under the K=5 keyframe setting. We evaluate five metrics. Arrows indicate the desired direction (bold is the best for error/quality metrics).

Our method consistently outperforms all methods. These results demonstrate that our model not only synthesize motions more precisely and follows target trajectories more faithfully, but also preserves the statistical fidelity of the generated motions to the real data, without compromising diversity.

For further comparison, Figure 2 presents qualitative results of our model against other methods. In Figure 4, the first two plots illustrate how MPJPE and FID vary as the number of keyframe conditions increases. The similar curve shapes and smooth trends indicate that our model adapts well to different conditioning scenarios. As expected, providing more keyframe poses consistently leads to higher-quality generations with lower metric values. The radar plot on the right summarizes the overall capability of different models, where our MoFA achieves the largest coverage, demonstrating superior comprehensive performance. More results could be seen in Figure 3 and Appendix.

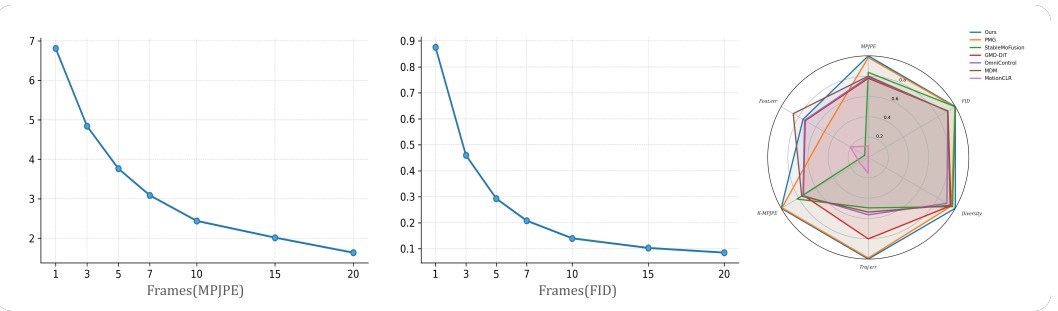

Figure 4: (Left) MPJPE decreases as the number of keyframe conditions increases, showing that more keyframe poses lead to more accurate reconstructions. (Middle) FID exhibits a similar decreasing trend, indicating improved realism with additional conditions. (Right) Radar chart comparing different methods across multiple metrics, where our MoFA achieves the largest overall area, demonstrating superior comprehensive performance.

### 4.1.2 COMPARED WITH BASE MODEL

To evaluate the effectiveness of our proposed motion factorization method, we additionally train a baseline model (**Base**) that is optimized only on high-quality datasets, without the QADT strategy and the use of additional low-quality data.

| Method | FS(cm) ↓ | JS ↓ | MF → |
|--------|----------|------|------|
| Base-1 | 3.544 | 0.018 | 0.824 |
| Base-3 | 3.385 | 0.018 | 0.828 |
| Base-5 | 3.484 | 0.019 | 0.825 |
| Base-7 | 3.563 | 0.020 | 0.816 |
| Base-10 | 3.851 | 0.022 | 0.806 |
| MoFA-1 | 2.953 | 0.017 | 0.812 |
| MoFA-3 | 2.990 | 0.017 | 0.813 |
| MoFA-5 | 3.085 | 0.018 | 0.808 |
| MoFA-7 | 3.334 | 0.020 | 0.797 |
| MoFA-10 | 3.442 | 0.021 | 0.788 |

Table 2: This table contains three metrics: foot sliding, joint smoothness, and motion autocorrelation, comparing the values of our model and the baseline model under different keyframe conditions. As can be seen from the table, our model achieves better results in each set of keyframe comparisons.

Table 3 compares this baseline (Base-*) with our method (MoFA-*) under varying numbers of conditioning keyframes $K \in \{1, 3, 5, 7, 10\}$, the trajectories input are same. We report the same five metrics (mean ± standard deviation) as introduced earlier. Across all $K$, **MoFA** consistently achieves better accuracy and distributional fidelity than the baseline. At K=5, MoFA achieves relative reductions of 5-6% for MPJPE; at K=10, FID improves by nearly 15%.

Trajectory error also benefits from our approach across all conditions, regardless of whether few or many keyframes are given. Importantly, MoFA maintains consistently low errors across the board. Motion diversity remains stable in the range of $\approx$ 4.48-4.56 and is comparable between methods; MoFA is slightly higher for K=1,5 and slightly lower at K=3,7,10, with overall differences under 0.03. This indicates that the accuracy gains do not come at the expense of motion variety. Furthermore, the reported standard deviations are modest, suggesting that the improvements are stable across runs. Overall, the results show that while both models improve as more keyframes are provided, MoFA leverages these additional constraints more effectively, producing lower errors while preserving diversity.

We further compare basline and MoFA on three perceptual metrics: foot sliding(FS), motion smoothness(JS), and motion autocorrelation (Table 2). MoFA achieves consistently lower values across all $K$, which we attribute to both the model's structure and the inclusion of a smoothness loss. On average, MoFA also improves upon baseline in terms of smoothness and autocorrelation.

### 4.2 QUALITATIVE DISCUSSION OF MOTION FACTORIZATION METHOD

In this section, we analyze from a qualitative perspective why our model assumptions are valid. Figure 5 shows the baseline and MoFA in feature level.

**PCA Features Analysis**: The left subfigure projects sequence-level local motions onto the first two principal components, comparing three settings. Under identical keyframe and trajectory conditions, our MoFA and baseline models (with LMG and GMG, respectively) generate both global and local motion sequences. For global motions, we further separate them into local components, yielding

| Method | MPJPE ↓ | FID ↓ | Diversity → | Traj.err ↓ | K-MPJPE ↓ |
|---|---|---|---|---|---|
| Base-1 | 7.045 ±.104 | 0.995 ±.033 | 4.478 ±.012 | 0.099 ±.028 | 7.037 ±.100 |
| Base-3 | 5.035 ±.075 | 0.513 ±.020 | 4.508 ±.014 | 0.110 ±.026 | 4.474 ±.074 |
| Base-5 | 3.983 ±.077 | 0.331 ±.019 | 4.544 ±.018 | 0.084 ±.020 | 3.344 ±.061 |
| Base-7 | 3.264 ±.052 | 0.245 ±.015 | 4.543 ±.014 | 0.137 ±.065 | 2.744 ±.041 |
| Base-10 | 2.619 ±.034 | 0.165 ±.010 | 4.546 ±.014 | 0.124 ±.061 | 2.173 ±.033 |
| MoFA-1 | 6.809 ±.143 | 0.876 ±.042 | 4.482 ±.017 | 0.080 ±.023 | 6.798 ±.141 |
| MoFA-3 | 4.838 ±.086 | 0.460 ±.022 | 4.505 ±.015 | 0.072 ±.017 | 4.308 ±.077 |
| MoFA-5 | 3.763 ±.083 | 0.293 ±.016 | 4.556 ±.012 | 0.078 ±.013 | 3.163 ±.072 |
| MoFA-7 | 3.088 ±.067 | 0.208 ±.011 | 4.525 ±.015 | 0.118 ±.035 | 2.620 ±.056 |
| MoFA-10 | 2.438 ±.051 | 0.140 ±.007 | 4.535 ±.018 | 0.073 ±.020 | 2.015 ±.038 |

Table 3: Comparisons between our model and baseline. This table shows at different keyframes situations, our MoFA with QADT are always better than the baseline.

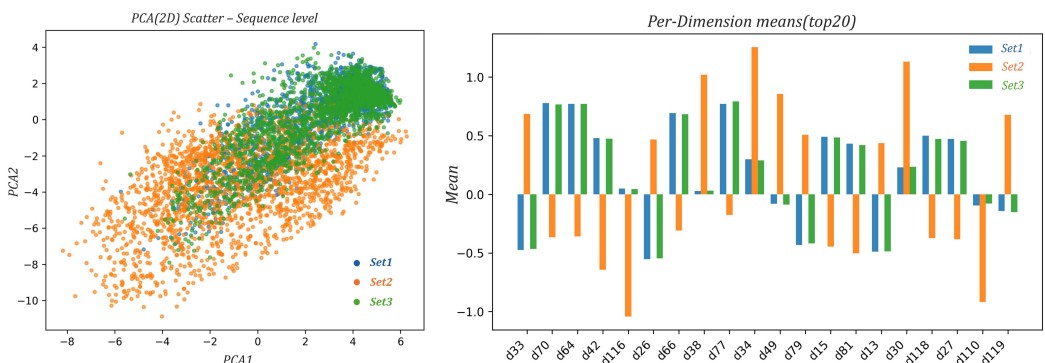

Figure 5: The figure visualizes the feature distribution of our MoFA and baseline, to demonstrate the effectiveness of Motion Factorization method proposed.

Set1 (MoFA's local motions from global output) and Set3 (baseline's local motions from global output). For local motions, the trajectory is treated as the default parametric variable learned by the model, yielding Set2 (MoFA).

The visualization reveals two key findings: (i) trajectory conditioning sharpens distributions—Set1 (blue) and Set2 (orange) share similar geometry shape, but Set1 is denser, showing that explicit trajectory input reduces stochasticity and aligns motions with the principal manifold; (ii) QADT and LMRS cause no domain shift—Set3 (green) overlaps with Set1, confirming that additional local-motion data and branches do not affect the weight domain while providing a stable basis for convergence.

**Embedding Analysis**: The bar chart shows the mean of the top 20 embedding dimensions with largest between-set deviations (Set1, Set2, Set3), highlighting systematic feature shifts. Two patterns emerge: (i) Set1 and Set3 align closely, with overlapping directions and magnitudes across most dimensions, indicating that MoFA and the baseline share latent regions and that QADT does not induce domain shift when incorporating additional local-motion data. (ii) Set2 deviates consistently, with larger magnitudes and frequent sign flips (e.g., negative bias at d116, positive shifts at d34, d30, d118), suggesting that QADT yields a broader and more ambiguous local-motion feature domain.

Overall, per-dimension means show that local motions from MoFA and the baseline share stable feature distributions, while QADT, despite shifting means, preserves alignment with the base model and improves convergence. Together with PCA and bar chart analyses, these results confirm that trajectory inputs stabilize local-motion distributions, and QADT regularizes the feature domain without degrading model capacity. Therefore we demonstrate our method's assumption is reasonable and correct.

### 4.3 IMPACT OF FOOT-CONTACT LOSS ON PHYSICAL REALISM

While previous evaluations based on MPJPE, FID, and feature-space metrics demonstrate the advantages of our model, human motion generation ultimately serves practical and realistic applications, where the physical plausibility of the generated motion is crucial. To further examine whether our method produces physically grounded motion rather than simply performing well on evaluation metrics, we introduce a foot-contact loss in this section. This addition does not modify the overall training pipeline; instead, it imposes a lightweight physical constraint on foot movement. Our goal is to assess whether the model can achieve more physically realistic generation under this constraint.

To encourage physically plausible foot-ground interactions, we introduce a foot-contact loss that penalizes non-zero foot velocities during contact frames. Contact frames are detected from ground-truth motion based on near-static foot velocities and converted into binary masks to constrain the predicted joint trajectories. When a frame is marked as contact, the predicted foot velocity is encouraged toward zero, effectively reducing sliding artifacts and improving motion realism without altering the model architecture.

| Method | FS(cm) ↓ | JS ↓ | MF → | Method | FS(cm) ↓ | JS ↓ | MF → |
|---|---|---|---|---|---|---|---|
| MDM-5 | **3.815** | **0.006** | **0.916** | MDM-10 | **3.853** | **0.006** | **0.914** |
| StableMoFusion-5 | 3.844 | 0.013 | 0.785 | StableMoFusion-10 | 3.862 | 0.013 | 0.781 |
| PMG-5 | 3.816 | 0.009 | 0.841 | PMG-10 | 3.893 | 0.011 | 0.830 |
| MoFA-5 | 3.947 | 0.011 | 0.852 | MoFA-10 | 3.961 | 0.014 | 0.820 |

Table 4: The table reports Foot Sliding (FS), Joint Smoothness (JS), and Motion Fluency (MF) for four models (MDM, StableMoFusion, PMG, MoFA) under two guidance settings (5 keyframes and 10 keyframes), without using foot-contact loss.

As shown in Table 4, MDM obtains the best scores across all metrics and under both keyframe conditions. However, this does not necessarily indicate inherent superiority; rather, MDM tends to produce more conservative motions with smaller amplitude, which naturally reduces accumulated foot sliding and increases smoothness. Furthermore, the small differences among the four models in FS, JS, and MF suggest that our MoFA does not introduce noticeable or systematic foot-sliding artifacts, confirming that the proposed framework does not bias the generation toward unstable contact or discontinuous motion.

| Method | FS(cm) ↓ | JS ↓ | MF → | Method | FS(cm) ↓ | JS ↓ | MF → |
|---|---|---|---|---|---|---|---|
| MDM-5 | **0.557** | **0.003** | **0.890** | MDM-10 | 0.565 | **0.003** | **0.895** |
| MoFA-5 | 0.558 | 0.004 | 0.781 | MoFA-10 | **0.562** | 0.005 | 0.747 |

Table 5: The table reports Foot Sliding (FS), Joint Smoothness (JS) and Motion Fluency (MF) for MDM and MoFA when contact loss is applied (using the same formulation as in MDM). We evaluate both models under 5-keyframe and 10-keyframe guidance.

Compared with Table 5 (with contact loss), both models show a substantial reduction in FS, confirming the effectiveness of the contact loss in mitigating foot-ground drifting. Notably, our model adapts to this constraint without architectural changes and remains comparable to MDM across all three metrics under both guidance settings.

## 5    CONCLUSION

In this work, we proposed **MoFA**, a diffusion-based Motion Factorization framework that explicitly disentangles local keyframe alignment from global trajectory consistency. By decomposing motion synthesis into complementary sub-tasks—local motion completion and trajectory adaptation— MoFA alleviates the entanglement between keyframe and trajectory signals that often leads to overfitting and poor generalization in existing approaches. Our framework integrates two modules, LMRS and TAMI, to unify local refinement with global adaptation, and is further enhanced by a Quality-Aware Dual Training (QADT) strategy that leverages both high-quality and imperfect data. Together, these components enable MoFA to generate precise and stable motions under limited, noisy, or user-defined conditions, while maintaining robustness and scalability.

Extensive experiments validate that MoFA not only improves controllability and distributional stability but also achieves superior generalization compared to existing baselines. Quantitative and qualitative analysis also proved the effectiveness of our method.

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

# A APPENDIX

## A.1 THE USE OF LARGE LANGUAGE MODELS (LLMS)

In this work, the LLM plays the role of polishing the grammar and words of the paper.

## A.2 NETWORK ARCHITECTURE

Our **MoFA** framework is realized as a transformer-based diffusion model that explicitly factorizes motion synthesis into two complementary branches: the *Local Motion Refinement Stack (LMRS)* and the *Trajectory-Aware Motion Integration (TAMI)*. This modular design allows MoFA to first refine local motion features and subsequently adapt them to global trajectory constraints.

**Linear Block and Condition Encoder.** The input signals consist of the diffusion timestep $t$, trajectory information, and keyframe constraints. Each input is processed by its dedicated encoder: the timestep $t$ is embedded using a linear block, trajectories are encoded by a lightweight MLP, and keyframes are projected through a linear embedding layer. All embeddings are mapped into a hidden representation with dimension $d = 512$, ensuring a uniform latent space for subsequent integration.

**Constraint Integration Module (CIM).** The CIM receives two condition embeddings—keyframe and trajectory—and fuses them into a unified representation. Fusion is realized through a sequence of self-attention and cross-attention layers with hidden size $d = 512$, enabling CIM to capture both intra-condition dependencies and inter-condition correlations. The resulting embedding serves as a refined local keyframe representation, which is then passed to the LMRS for further processing.

**Local Motion Refinement Stack (LMRS).** LMRS is designed to synthesize plausible local motion sequences directly from keyframe cues. It consists of $L = 8$ stacked transformer blocks, each with hidden size $d = 512$ and 8 attention heads. LMRS refines the local keyframe embeddings produced by CIM, generating temporally coherent local motion features that capture short-range dependencies and fine-grained pose dynamics.

**Trajectory-Aware Motion Integration (TAMI).** While LMRS focuses on local motion coherence, TAMI enforces global spatial consistency by aligning motions with trajectory inputs. Trajectories are first encoded by a MLP into a trajectory embedding, which is then fused with the LMRS outputs through cross-attention. This integration ensures that the generated motions faithfully follow the prescribed spatial path, preventing drift or misalignment between trajectory and local pose dynamics.

**Diffusion Heads.** Unlike conventional diffusion models that directly predict Gaussian noise, MoFA employs two specialized generators: the *Local Motion Generator (LMG)* attached to LMRS and the *Global Motion Generator (GMG)* attached to TAMI. Both generators are implemented as linear projections that output motion representations in a realistic parametric format. This design provides more explicit supervision than noise, enabling stronger gradient signals and facilitating stable convergence.

**Training Details.** We adopt a diffusion horizon of 1000 steps with a cosine noise schedule. Both LMRS and TAMI are jointly optimized under the proposed *Quality-Aware Dual Training (QADT)* strategy. High-quality datasets supervise both local and global branches, while low-quality dataset only go through local motion brach. This dual-training scheme expands the effective training set, mitigates overfitting, and enhances the model's generalization capability.

## A.3 TRAINING SETUP

We adopt the AdamW optimizer with a weight decay of $1 \times 10^{-6}$, combined with an exponential moving average (EMA) of the model parameters. The learning rate is scheduled using OneCycleLR with a maximum value of $1 \times 10^{-4}$. The model is trained with a batch size of 128 on an NVIDIA RTX 4070Ti GPU.

### A.4 Data Processing

In data processing, we adopt a unified representation to ensure compatibility between high-quality and low-quality datasets, enabling the model to handle both within a single framework. Specifically, for the high-quality dataset, each motion frame is represented as $f_1 \in \mathbb{R}^{132}$: the first 132 dimensions encode rotations of 22 joints in 6D format, the next 6 dimensions encode the global trajectory. For the low-quality dataset, we preserve the same structure but omit the unreliable global trajectory, resulting in a representation $f_2 \in \mathbb{R}^{138}$ that includes 132 dimensions for joint rotations. Thus, $f_2 - f_1 = 6$, with the only difference being the trajectory information. This design ensures a consistent feature format across datasets and guarantees compatibility in model training.

### A.5 Trainnig Strategy

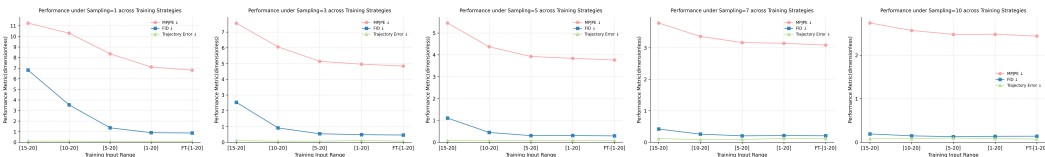

Figure 6: The figure illustrates model performance across different training stages under five keyframe settings1,3,5,7,10. As training stages, the metrics—MPJPE, FID, and trajectory error—consistently improve, indicating enhanced accuracy, realism, and trajectory fidelity.

To facilitate stable convergence, we trained the model for 100 epochs using a curriculum learning strategy based on progressive keyframe sampling. Specifically, the training process was divided into four stages, each lasting 20 epochs, where the keyframe selection interval was gradually narrowed from $[15, 20]$ to $[10, 20]$, then $[5, 20]$, and finally $[1, 20]$. This design creates a learning trajectory from easier to more challenging tasks, enabling the model to gradually adapt to sparser supervision. In the final stage, we fine-tuned the model for an additional 20 epochs on the $[1, 20]$ interval with a reduced learning rate of $1 \times 10^{-6}$. This curriculum schedule promotes more stable convergence and improves the quality of synthesized motions. The qualitative result could be shown in Figure6.

### A.6 More quantitative experiments

In this section, we will list more tables to show the comparison of indicators between different models.

i) Table 6 reports a quantitative comparison between our method and recent state-of-the-art baselines across five metrics: MPJPE, FID, Diversity, Trajectory Error, and K-MPJPE. Our approach (Ours-1) achieves the best performance on all metrics, yielding the lowest MPJPE (6.809), FID (0.876), and trajectory error (0.080), while also attaining the highest diversity (4.482) and the lowest K-MPJPE (6.798).

| Method | MPJPE ↓ | FID ↓ | Diversity → | Traj.err ↓ | K-MPJPE ↓ |
|---|---|---|---|---|---|
| CAMDM-1(Chen et al., 2024b) | 7.937 ±.145 | 2.064 ±.058 | 4.196 ±.019 | 0.523 ±.136 | 7.877 ±.140 |
| MDM-1(Tevet et al., 2022b) | 7.734 ±.094 | 1.432 ±.044 | 4.394 ±.018 | 0.589 ±.109 | 7.711 ±.089 |
| GMD-Unet-1(Guo et al., 2025) | 11.871 ±.091 | 3.009 ±.075 | 4.425 ±.013 | 0.728 ±.077 | 11.926 ±.125 |
| GMD-Dit-1(Guo et al., 2025) | 7.941 ±.126 | 1.208 ±.029 | 4.440 ±.019 | 0.172 ±.021 | 7.914 ±.125 |
| MotionCLR-1(Chen et al., 2024a) | 11.002 ±.116 | 5.410 ±.128 | 3.798 ±.018 | 0.641 ±.099 | 11.039 ±.128 |
| MotionDiffuse-1(Dabral et al., 2023) | 7.847 ±.122 | 0.965 ±.034 | 4.474 ±.019 | 0.598 ±.088 | 7.858 ±.112 |
| PriorMDM-1(Shafir et al., 2023) | 9.922 ±.130 | 1.939 ±.071 | 4.360 ±.017 | 0.566 ±.159 | 9.926 ±.133 |
| StableMoFusion-1(Huang et al., 2024) | 7.762 ±.137 | 0.884 ±.013 | 4.477 ±.014 | 0.482 ±.094 | 7.759 ±.143 |
| OmniControl-1(Xie et al., 2024) | 7.417 ±.143 | 1.366 ±.048 | 4.368 ±.016 | 0.392 ±.071 | 7.419 ±.146 |
| PMG-1(Xi et al., 2025) | 6.896 ±.011 | 1.141 ±.037 | 4.408 ±.016 | 0.152 ±.059 | 6.883 ±.125 |
| Ours-1 | **6.809** ±.143 | **0.876** ±.042 | **4.482** ±.017 | **0.080** ±.023 | **6.798** ±.141 |

Table 6: Under the one-keyframe sampling situation, the quantitative comparison with advanced methods on multiple metrics, including MPJPE, FID, Diversity, Trajectory error, and K-MPJPE. The best results are highlighted in bold.

ii) Table 7 presents quantitative comparisons across five metrics: MPJPE, FID, Diversity, Trajectory Error, and K-MPJPE, under the 3-keyframe setting. Our method (Ours-3) consistently outperforms all baselines, achieving the lowest MPJPE (4.838), FID (0.460), and trajectory error (0.072), while also delivering the best K-MPJPE (4.308) and competitive diversity (4.505).

| Method | MPJPE ↓ | FID ↓ | Diversity → | Traj.err ↓ | K-MPJPE ↓ |
|---|---|---|---|---|---|
| CAMDM-3(Chen et al., 2024b) | 6.130 ±.100 | 1.096 ±.031 | 4.415 ±.016 | 0.572 ±.141 | 5.778 ±.087 |
| MDM-3(Tevet et al., 2022b) | 6.084 ±.085 | 0.879 ±.026 | 4.480 ±.018 | 0.496 ±.013 | 5.740 ±.090 |
| GMD-Unet-3(Guo et al., 2025) | 11.933 ±.101 | 3.034 ±.054 | 4.430 ±.011 | 0.750 ±.088 | 11.779 ±.096 |
| GMD-Dit-3(Guo et al., 2025) | 6.347 ±.101 | 0.873 ±.026 | 4.498 ±.016 | 0.208 ±.050 | 6.004 ±.098 |
| MotionCLR-3(Chen et al., 2024a) | 11.011 ±.130 | 5.461 ±.093 | 3.790 ±.020 | 0.722 ±.127 | 10.916 ±.130 |
| MotionDiffuse-3(Dabral et al., 2023) | 6.429 ±.079 | 0.665 ±.024 | 4.504 ±.013 | 0.534 ±.095 | 6.157 ±.071 |
| PriorMDM-3(Shafir et al., 2023) | 8.500 ±.100 | 1.635 ±.079 | 4.355 ±.019 | 0.619 ±.123 | 8.358 ±.101 |
| StableMoFusion-3(Huang et al., 2024) | 5.785 ±.102 | 0.473 ±.022 | **4.524** ±.015 | 0.403 ±.096 | 5.306 ±.085 |
| OmniControl-3(Xie et al., 2024) | 6.115 ±.077 | 0.891 ±.029 | 4.441 ±.014 | 0.363 ±.052 | 5.802 ±.070 |
| PMG-3(Xi et al., 2025) | 4.961 ±.074 | 0.542 ±.024 | 4.481 ±.013 | 0.150 ±.065 | 4.360 ±.064 |
| Ours-3 | **4.838** ±.086 | **0.460** ±.022 | 4.505 ±.015 | **0.072** ±.017 | **4.308** ±.077 |

Table 7: Under the three-keyframe sampling situation, the quantitative comparison with advanced methods on multiple metrics, including MPJPE, FID, Diversity, Trajectory error, and K-MPJPE. The best results are highlighted in bold.

iii) Table 8 presents quantitative comparisons across five metrics: MPJPE, FID, Diversity, Trajectory Error, and K-MPJPE, under the 7-keyframe setting. Our method (Ours-7) consistently outperforms all baselines, achieving the lowest MPJPE (3.088), FID (0.208), and trajectory error (0.118), while also delivering the best K-MPJPE (2.620) and competitive diversity (4.525).

| Method | MPJPE ↓ | FID ↓ | Diversity → | Traj.err ↓ | K-MPJPE ↓ |
|---|---|---|---|---|---|
| CAMDM-7(Chen et al., 2024b) | 5.147 ±.099 | 0.845 ±.033 | 4.483 ±.018 | 0.428 ±.096 | 5.062 ±.103 |
| MDM-7(Tevet et al., 2022b) | 5.073 ±.077 | 0.692 ±.020 | 4.514 ±.011 | 0.412 ±.076 | 4.985 ±.072 |
| GMD-Unet-7(Guo et al., 2025) | 11.885 ±.119 | 3.021 ±.070 | 4.431 ±.013 | 0.746 ±.081 | 11.777 ±.121 |
| GMD-Dit-7(Guo et al., 2025) | 5.211 ±.071 | 0.632 ±.022 | 4.524 ±.017 | 0.282 ±.094 | 5.109 ±.065 |
| MotionCLR-7(Chen et al., 2024a) | 11.028 ±.097 | 5.442 ±.162 | 3.801 ±.014 | 0.688 ±.103 | 10.956 ±.091 |
| MotionDiffuse-7(Dabral et al., 2023) | 5.626 ±.117 | 0.541 ±.028 | **4.546** ±.017 | 0.396 ±.056 | 5.559 ±.115 |
| PriorMDM-7(Shafir et al., 2023) | 7.616 ±.105 | 1.518 ±.033 | 4.386 ±.020 | 0.635 ±.121 | 7.557 ±.103 |
| StableMoFusion-7(Huang et al., 2024) | 4.820 ±.084 | 0.235 ±.015 | 4.543 ±.015 | 0.429 ±.114 | 4.574 ±.078 |
| OmniControl-7(Xie et al., 2024) | 5.229 ±.103 | 0.688 ±.024 | 4.484 ±.015 | 0.364 ±.059 | 5.166 ±.097 |
| PMG-7(Xi et al., 2025) | 3.202 ±.071 | 0.224 ±.010 | 4.519 ±.016 | 0.119 ±.052 | 2.690 ±.055 |
| Ours-7 | **3.088** ±.067 | **0.208** ±.011 | 4.525 ±.015 | **0.118** ±.035 | **2.620** ±.056 |

Table 8: Under the seven-keyframe sampling situation, the quantitative comparison with advanced methods on multiple metrics, including MPJPE, FID, Diversity, Trajectory error, and K-MPJPE. The best results are highlighted in bold.

iv) Table 9 presents quantitative comparisons across five metrics: MPJPE, FID, Diversity, Trajectory Error, and K-MPJPE, under the 10-keyframe setting. Our method (Ours-10) consistently outperforms all baselines, achieving the lowest MPJPE (2.438), FID (0.140), and trajectory error (0.073), while also delivering the best K-MPJPE (2.015) and competitive diversity (4.535).

| Method | MPJPE ↓ | FID ↓ | Diversity → | Traj.err ↓ | K-MPJPE ↓ |
|---|---|---|---|---|---|
| CAMDM-10(Chen et al., 2024b) | 4.974 ±.085 | 0.815 ±.033 | 4.486 ±.017 | 0.462 ±.101 | 4.933 ±.083 |
| MDM-10(Tevet et al., 2022b) | 4.852 ±.095 | 0.664 ±.023 | 4.537 ±.014 | 0.444 ±.099 | 4.815 ±.093 |
| GMD-Unet-10(Guo et al., 2025) | 11.880 ±.128 | 3.020 ±.071 | 4.429 ±.015 | 0.791 ±.085 | 11.807 ±.128 |
| GMD-Dit-10(Guo et al., 2025) | 4.904 ±.081 | 0.569 ±.023 | 4.526 ±.018 | 0.184 ±.052 | 4.854 ±.080 |
| MotionCLR-10(Chen et al., 2024a) | 11.035 ±.092 | 5.476 ±.168 | 3.793 ±.020 | 0.701 ±.093 | 10.971 ±.091 |
| MotionDiffuse-10(Dabral et al., 2023) | 5.433 ±.094 | 0.503 ±.016 | **4.544** ±.015 | 0.395 ±.080 | 5.399 ±.094 |
| PriorMDM-10(Shafir et al., 2023) | 7.373 ±.112 | 1.473 ±.049 | 4.409 ±.016 | 0.575 ±.188 | 7.335 ±.113 |
| StableMoFusion-10(Huang et al., 2024) | 4.529 ±.097 | 0.202 ±.011 | 4.542 ±.014 | 0.334 ±.117 | 4.359 ±.091 |
| OmniControl-10(Xie et al., 2024) | 5.061 ±.098 | 0.660 ±.025 | 4.493 ±.021 | 0.305 ±.036 | 5.025 ±.095 |
| PMG-10(Xi et al., 2025) | 2.552 ±.062 | 0.163 ±.010 | 4.541 ±.017 | 0.085 ±.022 | 2.163 ±.0494 |
| Ours-10 | **2.438** ±.051 | **0.140** ±.007 | 4.535 ±.018 | **0.073** ±.020 | **2.015** ±.038 |

Table 9: Under the ten-keyframe sampling situation, the quantitative comparison with advanced methods on multiple metrics, including MPJPE, FID, Diversity, Trajectory error, and K-MPJPE. The best results are highlighted in bold.

v) Ablation, in Table we evaluate the effects of the module CIM, LMRS and TAMI. For CIM, we replace it by simple add operation, For LMRS, we remove it, and for TAMI, we replace it by simple add operation. Table10 reports the ablation study evaluating individual components and the full model. The results show that the complete model consistently achieves the best performance across different sampling situations, underscoring the effectiveness of the three key components: CIM, LMRS, and TAMI.

| Method | MPJPE ↓ | FID ↓ | Diversity → | Traj.err ↓ | K-MPJPE ↓ |
|---|---|---|---|---|---|
| w/o CIM | 7.642 ±.164 | 1.152 ±.053 | 4.311 ±.020 | 0.094 ±.029 | 7.601 ±.158 |
| w/o LMRS | 8.476 ±.187 | 1.482 ±.061 | 4.205 ±.024 | 0.098 ±.034 | 8.132 ±.179 |
| w/o TAMI | 7.588 ±.159 | 1.164 ±.055 | 4.326 ±.021 | 0.109 ±.028 | 7.454 ±.162 |
| Ours-1 | **6.809** ±.143 | **0.876** ±.042 | **4.482** ±.017 | **0.080** ±.023 | **6.798** ±.141 |
| w/o CIM | 5.012 ±.090 | 0.505 ±.025 | 4.476 ±.018 | 0.084 ±.018 | 4.882 ±.082 |
| w/o LMRS | 5.421 ±.097 | 0.612 ±.030 | 4.390 ±.020 | 0.109 ±.021 | 5.171 ±.089 |
| w/o TAMI | 5.048 ±.091 | 0.498 ±.024 | 4.469 ±.017 | 0.085 ±.019 | 4.795 ±.084 |
| Ours-3 | **4.838** ±.086 | **0.460** ±.022 | **4.505** ±.015 | **0.072** ±.017 | **4.308** ±.077 |
| w/o CIM | 3.941 ±.091 | 0.327 ±.018 | 4.489 ±.015 | 0.089 ±.014 | 3.798 ±.079 |
| w/o LMRS | 4.156 ±.102 | 0.361 ±.021 | 4.402 ±.018 | 0.097 ±.017 | 3.856 ±.085 |
| w/o TAMI | 3.918 ±.088 | 0.333 ±.017 | 4.473 ±.014 | 0.091 ±.015 | 3.612 ±.081 |
| Ours-5 | **3.763** ±.083 | **0.293** ±.016 | **4.556** ±.012 | **0.078** ±.013 | **3.163** ±.072 |
| w/o CIM | 3.254 ±.072 | 0.239 ±.014 | 4.482 ±.018 | 0.092 ±.037 | 2.951 ±.061 |
| w/o LMRS | 3.487 ±.081 | 0.276 ±.016 | 4.421 ±.020 | 0.108 ±.041 | 3.233 ±.067 |
| w/o TAMI | 3.291 ±.070 | 0.242 ±.012 | 4.475 ±.017 | **0.087** ±.038 | 2.869 ±.059 |
| Ours-7 | **3.088** ±.067 | **0.208** ±.011 | **4.525** ±.015 | 0.118 ±.035 | **2.620** ±.056 |
| w/o CIM | 2.615 ±.062 | 0.178 ±.010 | 4.421 ±.024 | 0.095 ±.018 | 2.589 ±.047 |
| w/o LMRS | 2.884 ±.075 | 0.195 ±.013 | 4.210 ±.031 | 0.091 ±.025 | 2.637 ±.058 |
| w/o TAMI | 2.641 ±.059 | 0.183 ±.009 | 4.398 ±.020 | 0.097 ±.019 | 2.408 ±.051 |
| Ours-10 | **2.438** ±.051 | **0.140** ±.007 | **4.535** ±.018 | **0.073** ±.020 | **2.015** ±.038 |

Table 10: Under the 1,3,5,7,10-keyframe sampling situations, the quantitative comparison of ablation studies on multiple metrics, including MPJPE, FID, Diversity, Trajectory error, and K-MPJPE. The best results are highlighted in bold.

## B MORE VISUALIZED RESULTS

Figure7 presents additional qualitative motion synthesis results generated by our model under diverse keyframe and trajectory conditions. The visualizations demonstrate that MoFA produces coherent and realistic human motions across a wide range of scenarios. These results highlight the model's ability to maintain spatial consistency, generate natural pose transitions, and preserve motion fidelity under varying inputs.

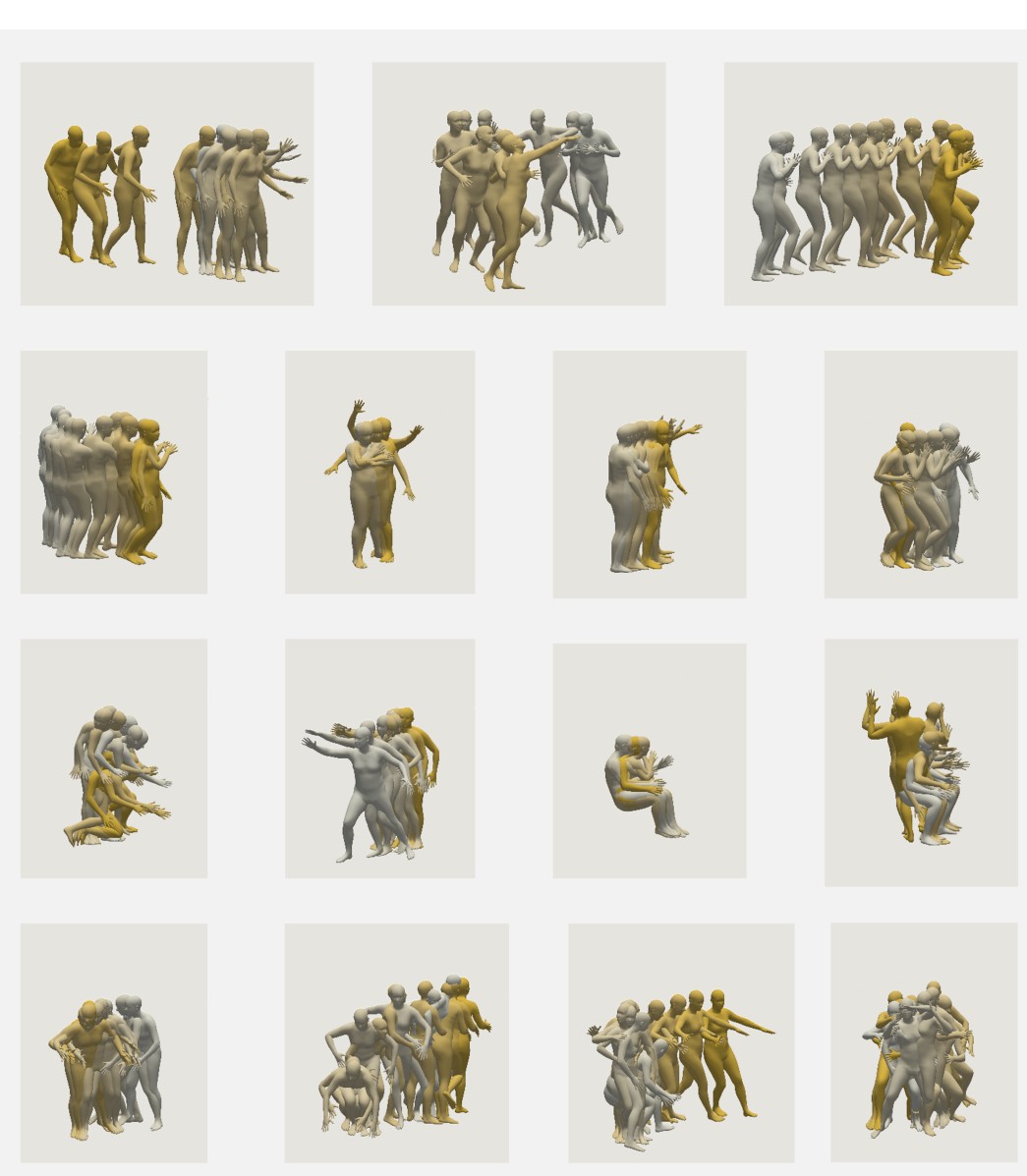

Figure 7: More visualized results.

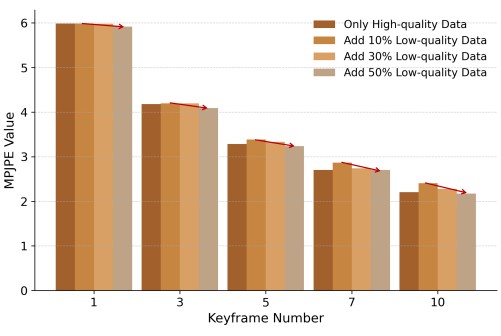 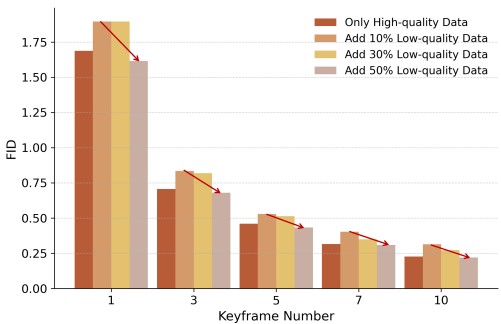

(a) Across different keyframe settings, MPJPE decreases as a larger proportion of low-quality data is introduced during training.

(b) Similarly, FID consistently decreases with increasing amounts of low-quality data, indicating improved generative realism.

Figure 8: Across all keyframe configurations, both MPJPE and FID follow the same trend in which incorporating more low-quality data gradually improves performance, with the 50% setting outperforming training on high-quality data alone.

## C  APPENDIX_REBUTTAL

- **Statement 1**: Given the limited rebuttal time, full-scale testing across all datasets could not be completed. To ensure a fair and meaningful supplementary evaluation, we report results on two representative datasets: HumanML3D ( 14k samples, high-quality) and a subset of MotionX++ ( 17k samples, lower-quality). We intentionally kept the data volume of both datasets comparable to enable rapid validation while still supporting reliable trend analysis.

### C.1  HOW DOES THE INCLUSION OF LOW-QUALITY DATA IMPROVE THE OVERALL PERFORMANCE OF THE METHOD?

Figure8 reports the quantitative influence of incorporating low-quality data during training, measured in terms of MPJPE and FID across different keyframe sampling conditions. The configuration Only High-quality Data denotes training exclusively on the clean dataset, whereas Add 10%/30%/50% Low-quality Data denotes joint training with progressively larger portions of noisy data. Across all five keyframe settings, we observe a consistent trend: models trained solely on high-quality data achieve better MPJPE and FID compared to those trained with a small amount of low-quality data. This illustrates that introducing 10% noisy data slightly deteriorates the data distribution and hinders the model's capacity to focus on the clean samples, resulting in worse performance. However, as the proportion of low-quality data increases, both MPJPE and FID steadily decline. This indicates that once the noisy data exceeds a certain threshold, it begins to provide complementary distributional information that enrich the model's generative ability rather than harm it. Remarkably, when 50% low-quality data is incorporated, both metrics surpass the baseline trained solely on high-quality data, demonstrating that abundant low-quality data does not degrade the model; instead, it can positively regularize training and ultimately enhance performance.

Figure9 presents the training loss curves over 100 epochs under four different data-combination settings. The green curve corresponds to training exclusively on high-quality data, for which the loss is computed only on clean samples. In contrast, the yellow, blue, and red curves jointly introduce 10%, 30%, and 50% low-quality data, respectively, and therefore optimize an additional loss term derived from the noisy portion of the dataset. As a result, their absolute loss values are necessarily higher than the green curve-not because the model performs worse, but simply because the optimization objective contains more components. The curves should therefore be interpreted primarily as indicators of convergence dynamics rather than final model quality. Notably, despite the higher absolute objective, incorporating low-quality data consistently accelerates convergence, and higher proportions of noisy samples lead to faster loss reduction and lower terminal training loss. The zoomed

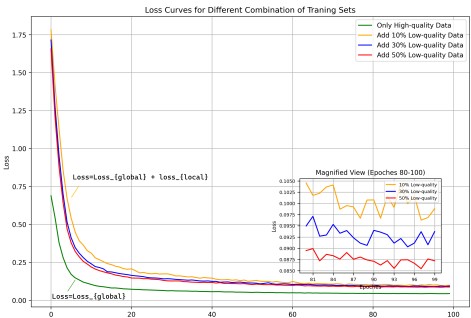

Figure 9: Training loss curves under different combinations of high-quality and low-quality data over 100 epochs, illustrating how increasing proportions of low-quality samples alter the overall optimization objective and convergence dynamics.

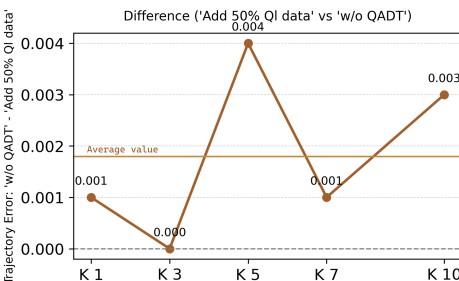

Figure 10: This figure illustrates the trajectory-error difference between training with 50% low-quality data and training solely on high-quality data across different keyframe settings.

view over epochs 80-100 further confirms this monotonic trend. These observations suggest that adding sufficient low-quality data enriches supervision in a way that benefits optimization.

Figure10 reports the trajectory-error differences between training solely on the high-quality dataset and training with an additional 50% low-quality data across different keyframe settings. Positive values indicate that models trained with 50% low-quality samples produce slightly higher trajectory errors than the high-quality-only baseline. While this degradation is consistently observable, its magnitude remains very small across all keyframes, with the difference fluctuating around an average of approximately $(2 \times 10^{-3})$. These results suggest that although incorporating low-quality data introduces a minor penalty in trajectory fidelity, the impact is negligible in practice and represents an acceptable trade-off when considering the substantial performance gains brought by low-quality data in other evaluation metrics.

Table11 summarizes the quantitative comparison between training solely on high-quality data (w/o QADT) and training with additional low-quality data at ratios of 10%, 30%, and 50% across different keyframe sampling configurations. A consistent trend emerges across all evaluation metrics (MPJPE, FID, Diversity, Traj.err, and K-MPJPE): the introduction of a small amount of low-quality data initially brings limited improvement, but progressively larger proportions of low-quality data lead to steady and noticeable performance gains. Notably, when 50% low-quality data is incorporated, the model achieves the best overall results across nearly all keyframe settings, surpassing the high-quality-only baseline. These results indicate that low-quality motion samples, when provided in sufficient quantity, enrich the training distribution and enhance both reconstruction accuracy and generative ability rather than hindering performance.

## C.2 How capable is the model itself?

In this section, we discuss the capabilities of the model itself.

| Method | MPJPE ↓ | FID ↓ | Diversity → | Traj.err ↓ | K-MPJPE ↓ |
|---|---|---|---|---|---|
| w/o QADT-1 | 5.988 ±.088 | 1.687 ±.046 | **4.309** ±.019 | **0.066** ±.004 | 5.890 ±.083 |
| 10% QADT-1 | 5.985 ±.153 | 1.896 ±.068 | 4.143 ±.022 | 0.079 ±.004 | 5.928 ±.144 |
| 30% QADT-1 | 5.982 ±.173 | 1.895 ±.065 | 4.213 ±.018 | 0.067 ±.005 | 5.899 ±.161 |
| 50% QADT-1 | **5.916** ±.133 | **1.615** ±.062 | 4.247 ±.024 | 0.067 ±.004 | **5.883** ±.150 |
| w/o QADT-3 | 4.182 ±.118 | 0.706 ±.036 | **4.412** ±.014 | 0.065 ±.003 | 3.698 ±.110 |
| 10% QADT-3 | 4.195 ±.086 | 0.834 ±.058 | 4.397 ±.023 | 0.073 ±.002 | 3.817 ±.090 |
| 30% QADT-3 | 4.196 ±.067 | 0.819 ±.043 | 4.328 ±.026 | 0.067 ±.002 | 3.759 ±.068 |
| 50% QADT-3 | **4.089** ±.107 | **0.679** ±.026 | 4.349 ±.021 | **0.065** ±.004 | **3.622** ±.101 |
| w/o QADT-5 | 3.285 ±.096 | 0.459 ±.028 | **4.416** ±.028 | **0.064** ±.005 | 2.848 ±.091 |
| 10% QADT-5 | 3.384 ±.081 | 0.528 ±.024 | 4.323 ±.015 | 0.074 ±.005 | 2.860 ±.074 |
| 30% QADT-5 | 3.333 ±.082 | 0.514 ±.025 | 4.382 ±.022 | 0.068 ±.003 | 2.852 ±.071 |
| 50% QADT-5 | **3.238** ±.089 | **0.433** ±.023 | 4.377 ±.014 | 0.068 ±.004 | **2.738** ±.081 |
| w/o QADT-7 | 2.704 ±.041 | 0.315 ±.015 | **4.432** ±.019 | **0.065** ±.004 | 2.347 ±.038 |
| 10% QADT-7 | 2.865 ±.058 | 0.403 ±.020 | 4.361 ±.023 | 0.076 ±.004 | 2.484 ±.051 |
| 30% QADT-7 | 2.740 ±.042 | 0.349 ±.012 | 4.383 ±.021 | 0.069 ±.003 | 2.358 ±.029 |
| 50% QADT-7 | **2.702** ±.091 | **0.309** ±.019 | 4.393 ±.016 | 0.066 ±.004 | **2.328** ±.077 |
| w/o QADT-10 | 2.204 ±.033 | 0.227 ±.011 | **4.431** ±.020 | **0.064** ±.003 | 1.901 ±.024 |
| 10% QADT-10 | 2.403 ±.086 | 0.314 ±.017 | 4.386 ±.021 | 0.073 ±.004 | 2.085 ±.031 |
| 30% QADT-10 | 2.279 ±.034 | 0.272 ±.014 | 4.405 ±.016 | 0.068 ±.003 | 1.951 ±.034 |
| 50% QADT-10 | **2.170** ±.028 | **0.220** ±.014 | 4.394 ±.020 | 0.067 ±.003 | **1.856** ±.025 |

Table 11: The table shows a comparison of various metrics under four scenarios: without using a QATD strategy, using QATD but only add 10%/30%/50% low-quality data. Each scenario contains 5 different situations of 1/3/5/7/10 key-frame conditions input.

| Method | MPJPE ↓ | FID ↓ | Diversity → | Traj.err ↓ | K-MPJPE ↓ |
|---|---|---|---|---|---|
| PMG-1 | 6.198 ±.117 | **1.594** ±.089 | 4.250 ±.012 | **0.065** ±.003 | 6.299 ±.125 |
| StableMoFusion-1 | 7.045 ±.149 | 1.855 ±.067 | 4.266 ±.019 | 0.261 ±.012 | 7.042 ±.139 |
| MoFA-1 | **5.988** ±.088 | 1.687 ±.046 | **4.309** ±.019 | 0.066 ±.004 | **5.890** ±.083 |
| PMG-3 | 4.484 ±.105 | 0.743 ±.032 | 4.334 ±.011 | 0.065 ±.002 | 3.989 ±.095 |
| StableMoFusion-3 | 5.125 ±.073 | 0.850 ±.051 | 4.307 ±.016 | 0.195 ±.009 | 4.784 ±.077 |
| MoFA-3 | **4.182** ±.118 | **0.706** ±.036 | **4.412** ±.014 | **0.065** ±.003 | **3.698** ±.110 |
| PMG-5 | 3.471 ±.069 | **0.438** ±.019 | 4.371 ±.017 | **0.062** ±.002 | 2.922 ±.060 |
| StableMoFusion-5 | 4.590 ±.092 | 0.574 ±.029 | 4.323 ±.023 | 0.165 ±.005 | 4.242 ±.089 |
| MoFA-5 | **3.285** ±.096 | 0.459 ±.028 | **4.416** ±.028 | 0.064 ±.005 | **2.848** ±.091 |
| PMG-7 | 2.874 ±.076 | **0.306** ±.024 | 4.395 ±.024 | **0.061** ±.001 | 2.423 ±.073 |
| StableMoFusion-7 | 4.327 ±.071 | 0.482 ±.025 | 4.428 ±.021 | 0.157 ±.007 | 4.116 ±.007 |
| MoFA-7 | **2.704** ±.041 | 0.315 ±.015 | **4.432** ±.019 | 0.065 ±.004 | **2.347** ±.038 |
| PMG-10 | 2.219 ±.070 | **0.195** ±.011 | 4.417 ±.024 | **0.062** ±.001 | 1.904 ±.052 |
| StableMoFusion-10 | 4.028 ±.102 | 0.326 ±.016 | 4.426 ±.026 | 0.142 ±.007 | 3.925 ±.097 |
| MoFA-10 | **2.204** ±.033 | 0.227 ±.011 | **4.431** ±.020 | 0.064 ±.003 | **1.901** ±.024 |

Table 12: This table compares our method with PMG and StableMotion when trained solely on high-quality data, highlighting the superior baseline performance of our approach across different keyframe settings.

### C.2.1 FAIR COMPARISON WITH OTHER MODELS

Table12 compares our model with two representative baselines, PMG and StableMotion, under the setting where all methods are trained solely on the high-quality dataset. These two baselines are chosen because they consistently outperform other competitors on average and therefore reflect a strong reference point for evaluating fundamental modeling capacity. Across all keyframe configurations and evaluation metrics, our method exhibits slightly superior base performance, indicating that the proposed architecture itself provides stronger reconstruction fidelity and generative quality even without leveraging additional supervisory signals. More importantly, unlike the baselines, our

| Method | MPJPE ↓ | FID ↓ | Diversity → | Traj.err ↓ | K-MPJPE ↓ |
|---|---|---|---|---|---|
| MoFA-FUSION-1 | 6.067 ±.116 | **1.561** ±.059 | **4.321** ±.018 | 0.070 ±.005 | 6.007 ±.107 |
| MoFA-1 | **5.988** ±.088 | 1.687 ±.046 | 4.309 ±.019 | **0.066** ±.004 | **5.890** ±.083 |
| MoFA-FUSION-3 | 4.353 ±.070 | **0.646** ±.029 | **4.438** ±.021 | 0.071 ±.004 | 3.928 ±.079 |
| MoFA-3 | **4.182** ±.118 | 0.706 ±.036 | 4.412 ±.014 | **0.065** ±.003 | **3.698** ±.110 |
| MoFA-FUSION-5 | 3.475 ±.076 | **0.385** ±.025 | **4.442** ±.022 | 0.069 ±.006 | 2.941 ±.055 |
| MoFA-5 | **3.285** ±.096 | 0.459 ±.028 | 4.416 ±.028 | **0.064** ±.005 | **2.848** ±.091 |
| MoFA-FUSION-7 | 2.904 ±.63 | **0.274** ±.012 | **4.463** ±.018 | 0.068 ±.005 | 2.491 ±.069 |
| MoFA-7 | **2.704** ±.041 | 0.315 ±.015 | 4.432 ±.019 | **0.065** ±.004 | **2.347** ±.038 |
| MoFA-FUSION-10 | 2.335 ±.070 | **0.208** ±.012 | **4.459** ±.016 | **0.062** ±.003 | 2.024 ±.059 |
| MoFA-10 | **2.204** ±.033 | 0.227 ±.011 | 4.431 ±.020 | 0.064 ±.003 | **1.901** ±.024 |

Table 13: This table compares our method with fusion version when trained solely on high-quality data, highlighting the superior baseline performance of our approach across different keyframe settings.

| Method | Params (M) | FLOPs (GF) | Latency(20fps,6s) | Sampling Steps |
|---|---|---|---|---|
| CAMDM | 26.05 | 1.06 | 1.316 | 35 |
| MDM | 25.97 | 1.04 | 0.711 | 35 |
| GMD-Unet | 232.42 | 9.36 | 1.946 | 35 |
| GMD-Dit | 39.10 | 1.05 | 0.721 | 35 |
| MotionCLR | 453.40 | 21.04 | 1.737 | 35 |
| MotionDiffuse | 85.44 | 2.86 | 1.180 | 35 |
| PriorMDM | 17.49 | 1.04 | 0.703 | 35 |
| OmniControl | 46.35 | 2.43 | 1.163 | 35 |
| StableMoFusion | 298.19 | 15.02 | 2.104 | 35 |
| PMG | 36.03 | 1.73 | 0.925 | 35 |
| MoFA | 52.84 | 2.93 | 2.011 | 35 |

Table 14: This table compares the computational efficiency of different motion generation models in terms of parameter size, FLOPs, inference latency, and sampling steps.

formulation is theoretically decomposable and can seamlessly accommodate joint training with low-quality motion data, which in subsequent experiments leads to substantial further performance gains as shown in Table11. This demonstrates that our model not only establishes a strong foundation but also uniquely benefits from the scalable incorporation of imperfect data.

### C.2.2 VERIFY THE VALIDITY OF THE CONDITIONAL INDEPENDENCE HYPOTHESIS.

Table15 evaluates the validity of the proposed independence assumption by comparing the original MoFA model with its variant MoFA-FUSION, where keyframe features are again injected into the global motion stage after the LMRS module. Both models are trained only on the high-quality dataset to ensure a controlled comparison. Across all keyframe settings, MoFA consistently achieves lower MPJPE and trajectory error than the FUSION variant, indicating that reintroducing keyframe signals at the global stage disrupts the decomposition and leads to larger accumulated pose drift. Interestingly, MoFA-FUSION yields slightly better FID, which reflects diversity rather than reconstruction stability. This trade-off suggests that MoFA behaves as a more conservative and reliability-oriented architecture: by enforcing a clean factorization and preventing the global stage from being influenced again by keyframe inputs, MoFA minimizes cumulative prediction errors and produces more stable motion, even at the cost of a modest reduction in output diversity. For human motion generation, such robustness and error resistance are generally more desirable than uncontrolled creativity, which further supports the semantic and empirical soundness of the proposed decomposition.

### C.3 MODEL INFERENCE RELATED INFORMATION.

The table14 reports the computational complexity and inference efficiency across representative motion generation models, including diffusion-based, transformer-based, and hybrid architectures. For

a fair comparison, latency is measured as the wall-clock time required to generate a single 6-second motion sequence at 20 fps (120 frames) with batch size 1 using DPM-Solver++ with 35 sampling steps. All results are obtained on an NVIDIA RTX 4070 Ti GPU in full FP32 precision. The results reveal a clear computational trade-off: larger models with higher FLOPs generally incur longer inference time, whereas lightweight architectures achieve faster sampling but may sacrifice model expressiveness. Notably, our model maintains competitive latency while possessing a moderate parameter count and FLOP budget, indicating that it achieves a favorable balance between computational cost and generative quality. This analysis emphasizes that efficiency is not merely determined by parameter size, but rather by how effectively the architecture translates computation into motion quality under a fixed sampling budget.

## C.4 EXPLANATION OF THE VISUAL VIDEO.

To further assess the behavior of the proposed framework under mismatched control conditions, we visualize four complementary scenarios.

Second, in 'same_traj_diff_keyf', identical trajectories combined with mismatched keyframes lead to results that keep the input local motion but differ in final trajectories. This is due to the injected trajectory is incorporated via cross-attention, the final motion trajectory is partially adapted by the keyframe condition, the fine-grained geometric details may shift. This behavior aligns with the goal of MoFA, which is to produce coherent global motions rather than strictly enforcing both signals even when they contradict.

Third, in 'direct_fusion_modals', directly combining mismatched trajectory and keyframe features causes noticeable collapse of global motion (e.g., characters drifting in place), demonstrating that naïve feature fusion cannot robustly cope with inconsistent control signals.

Finally, we include a representative 'bad case': a highly dynamic street-dance capture that lies outside the training distribution and contains rapid rotations, where MoFA exhibits unrealistic turning and ground sliding. This illustrates the current limitations of the model and suggests that incorporating additional domain-specific datasets (e.g., dance motion) is a promising direction for improving robustness.

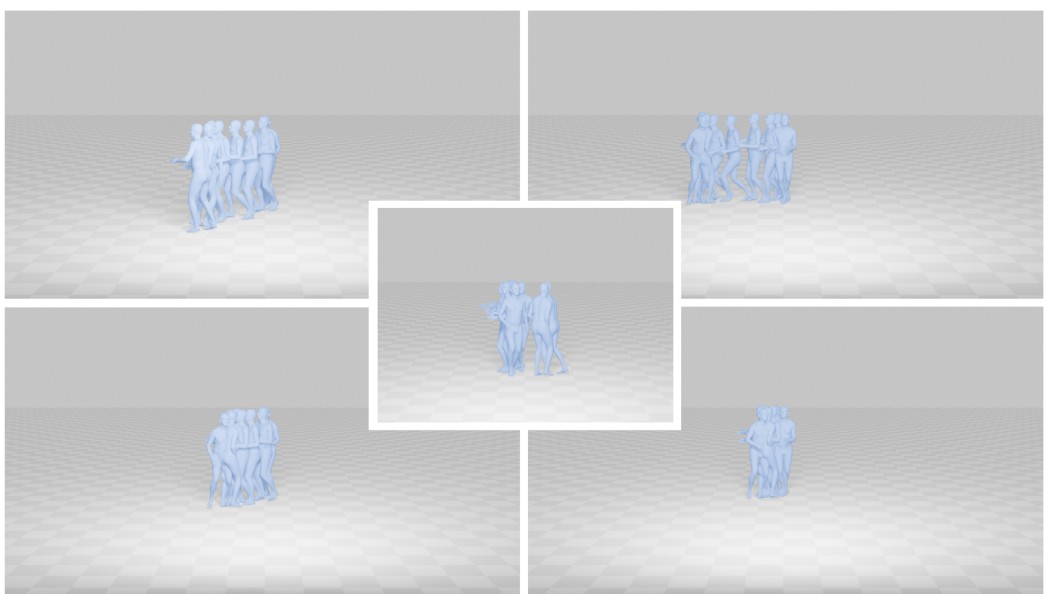

Figure 11: Visualization of generated motion showing body poses over time.

### C.5 REGARDING RENDERING QUALITY.

To avoid compilation timeout, we compressed each picture to under 10 MB, which unfortunately reduced the rendering sharpness of some visualizations. We fully agree that higher-resolution figures would better showcase the results. To address this, we have refined the rendering pipeline and produced improved high-resolution images, as illustrated in the updated example provided in Figure11. Since large-scale replacement of figures is discouraged during the rebuttal stage in order to preserve consistency of the submission, we will include all optimized renderings in the future releases.

### C.6 FOOT-CONTACT CONSTRAINTS.

To clearly explain the possible causes of foot-contact loss and our corresponding strategies, this section is divided into two parts: 1) the introduction fo the foot-related metrics and foot-contact loss function; 2) a comparison of all models under the same conditions (high-quality data only).

#### C.6.1 FURTHER EXPLANATION OF FOOT-RELATED METRICS AND CONTACT LOSS

- *For FID*. We follow the standard Fréchet distance protocol. For each real and generated motion sequence, we extract a 256-dimensional embedding using our pretrained motion encoder, whose parameters remain frozen during evaluation. We then compute the empirical mean and covariance of real and generated features separately and report the Fréchet distance values. The motion encoder's backbone is an 8-layer Transformer trained in a self-supervised contrastive manner on the motion dataset. Two stochastic augmentations (temporal masking and Gaussian noise) of the same motion form a positive pair, while other samples in the batch serve as negatives. Training follows an InfoNCE-style objective, encouraging invariance between augmented views of the same motion. After training, the encoder is used exclusively for feature extraction when computing FID.

- *For JS(joint smoothness)*. The JS metric evaluates the instantaneous physical smoothness of generated motion by computing the root-mean-square(RMS) joint acceleration. For each motion, we first convert 6D rotations to 3D joint positions, then compute joint velocities and accelerations via first- and second-order temporal differences. Lower JS values correspond to smaller, less abrupt changes in velocity, indicating smoother and more physically stable motion.

- *For MF(motion fluency). The MF metric evaluates temporal continuity by measuring the average autocorrelation of joint trajectories. After converting each sequence to joint positions and flattening them spatially, we compute the autocorrelation coefficient between the trajectory signal and a time-shifted version (lag = 5 frames). This coefficient is averaged across joints and samples. Higher MF values indicate more consistent temporal evolution and fewer unnecessary direction changes, reflecting more fluent and natural motion.*

  Importantly, JS and MF capture complementary aspects of motion dynamics: JS measures local smoothness by penalizing sudden accelerations, whereas MF measures global continuity by encouraging consistent temporal progression. They are not intended as absolute indicators of motion quality; a model could score well by generating overly conservative, low-amplitude motion. Their role is to quantify specific facets of physical plausibility, and they are interpreted in conjunction with other metrics such as FID and MPJPE.

- *For Foot Sliding*. The foot-sliding metric quantifies how much the feet undesirably move across the ground during expected contact phases. For each sequence, we accumulate horizontal foot displacement only when a foot is close to the ground (i.e., below a height threshold), and ignore displacement when the foot is lifted. The accumulated movement is averaged across frames and feet and converted to centimeters. Lower values indicate more stable foot-ground contact and fewer physically implausible sliding artifacts.

  In practice, JS and MF show very small variance (typically $< 1e\text{-}3$), as most generated motions are already highly smooth in terms of acceleration and temporal continuity. Foot-sliding, however, naturally exhibits much larger variance, because motions of different amplitude accumulate contact-phase displacement to very different degrees (small-range motions slide less, large-range motions slide more). Therefore, we mainly report the mean foot-sliding score, as its variance reflects motion type rather than model quality.

- *For Contact Loss.* To encourage physically plausible foot-ground interactions, we introduce a foot-contact loss that penalizes non-zero velocities of the feet during the ground-contact phase. Specifically, we detect contact frames from the ground-truth motion by identifying near-static velocities of foots, and use these binary contact masks to constrain the predicted joint trajectories. When a frame is labeled as contact, the predicted foot velocity is forced toward zero. This regularization reduces sliding artifacts and improves motion realism without modifying the model architecture.

### C.6.2 COMPARISON OF THE EFFECTS OF INTRODUCING FOOT CONTACT LOSS

According to the table 15, across all metrics and both keyframe conditions, MDM achieves the best scores. However, this does not imply that MDM is inherently superior. Rather, MDM tends to generate more conservative motions with smaller amplitude, which naturally results in lower accumulated foot sliding and higher smoothness/fluency. When interpreted together with FID and MPJPE results in the appendix, MDM does not outperform other methods in overall motion realism or precision.

| Method | FS(cm) ↓ | JS ↓ | MF → | Method | FS(cm) ↓ | JS ↓ | MF → |
|---|---|---|---|---|---|---|---|
| MDM-5 | **3.815** | **0.006** | **0.916** | MDM-10 | **3.853** | **0.006** | **0.914** |
| StableMoFusion-5 | 3.844 | 0.013 | 0.785 | StableMoFusion-10 | 3.862 | 0.013 | 0.781 |
| PMG-5 | 3.816 | 0.009 | 0.841 | PMG-10 | 3.893 | 0.011 | 0.830 |
| MoFA-5 | 3.947 | 0.011 | 0.852 | MoFA-10 | 3.961 | 0.014 | 0.820 |

Table 15: The table reports Foot Sliding (FS), Joint Smoothness (JS), and Motion Fluency (MF) for four models (MDM, StableMoFusion, PMG, MoFA) under two guidance settings (5 keyframes and 10 keyframes), without using foot-contact loss.

The differences among the four models in FS, JS, and MF are relatively small, indicating that our MoFA will not produce noticeable or systematic foot-sliding artifacts. These metrics therefore help confirm that the proposed framework does not unintentionally bias the generation toward unstable contact or discontinuous motion.

After introducing the foot-contact loss, the updated results are shown in the table16 and 17. Notably, this improvement does not require any modification to the training pipeline-only the addition of an extra loss term.

| Method | FS(cm) ↓ | JS ↓ | MF → | Method | FS(cm) ↓ | JS ↓ | MF → |
|---|---|---|---|---|---|---|---|
| MDM-5 | **0.557** | **0.003** | **0.890** | MDM-10 | 0.565 | **0.003** | **0.895** |
| MoFA-5 | 0.558 | 0.004 | 0.781 | MoFA-10 | **0.562** | 0.005 | 0.747 |

Table 16: The table reports Foot Sliding (FS), Joint Smoothness (JS) and Motion Fluency (MF) for MDM and MoFA when contact loss is applied (using the same formulation as in MDM). We evaluate both models under 5-keyframe and 10-keyframe guidance.

Compared with the table16 without contact loss, FS drops substantially for both models, indicating that contact loss is effective in reducing foot–ground drifting. Importantly, our model adapts to this constraint without architectural modification and remains close to MDM in all three metrics across both guidance settings. MDM still scores slightly better overall, which is consistent with the explanation provided earlier: MDM tends to generate more conservative, low-amplitude motions, naturally resulting in lower accumulated FS and higher smoothness/fluency.

We also observe that adding contact loss generally improves JS (less abrupt acceleration) while slightly decreasing MF (reduced long-range temporal continuity). This trade-off is intuitive: adding a physical constraint stabilizes ground contact and suppresses sudden movements, but also mildly restricts the temporal variability of motion. Crucially, the differences remain small, and both models produce stable contact and continuous motion, demonstrating that the proposed method does not introduce foot-sliding artifacts and is compatible with contact-based physical regularization.

To further explain how the foot-contact loss affects metrics such as FID and MPJPE. we report MPJPE, FID, Diversity, Traj.err, and K-MPJPE for MDM and MoFA under 5-keyframe and 10-keyframe guidance in table17, *without and with the contact loss (w/o C. and w/ C.).* Consistent with previous observations, the 10-keyframe setting generally outperforms the 5-keyframe setting, and MoFA achieves stronger overall accuracy than MDM across both. This confirms that the results shown here are aligned with the rest of our evaluation.

| Method | MPJPE ↓ | FID ↓ | Diversity → | Traj.err ↓ | K-MPJPE ↓ |
|---|---|---|---|---|---|
| MDM-5 w/o C. | 4.582 ±.084 | 1.133 ±.045 | **4.378** ±.026 | **0.346** ±.018 | 4.418 ±.083 |
| MDM-5 w/ C. | **1.887** ±.039 | **0.328** ±.024 | 3.004 ±.037 | 0.457 ±.019 | **1.825** ±.036 |
| MoFA-5 w/o C. | 3.285 ±.096 | 0.459 ±.028 | **4.416** ±.028 | **0.064** ±.005 | 2.848 ±.091 |
| MoFA-5 w/ C. | **1.690** ±.040 | **0.262** ±.019 | 2.973 ±.034 | 0.163 ±.006 | **1.458** ±.028 |
| MDM-10 w/o C. | 4.136 ±.059 | 1.064 ±.043 | **4.413** ±.021 | **0.329** ±.011 | 4.124 ±.057 |
| MDM-10 w/ C. | **1.738** ±.057 | **0.302** ±.022 | 3.011 ±.042 | 0.455 ±.016 | **1.733** ±.058 |
| MoFA-10 w/o C. | 2.204 ±.033 | 0.227 ±.011 | **4.431** ±.020 | **0.064** ±.003 | 1.901 ±.024 |
| MoFA-10 w/ C. | **1.152** ±.030 | **0.138** ±.010 | 2.938 ±.037 | 0.163 ±.007 | **1.050** ±.026 |

Table 17: The results demonstrate that adding the foot-contact loss significantly boosts motion quality (lower MPJPE, FID, K-MPJPE), while causing a minor decrease in Diversity and Traj.err.

Comparing *w/ contact loss vs. w/o contact loss* for the same model and same guidance setting, we observe substantial improvements in both MPJPE and FID, indicating that contact loss helps the model produce motions that are closer to the ground-truth distribution and reduces unrealistic details. At the same time, Diversity and Traj.err slightly decrease. This trade-off is reasonable: contact-based regularization reduces foot sliding and improves physical realism, which naturally narrows the range of lcoal motions and trajectories to lead to lower diversity, and makes the model less prone to "over-following" preplanned paths. Importantly, even with this reduction, the absolute values of Traj.err remains competitive and do not indicate a degradation of motion naturalness.

Overall, these results show that (1) introducing contact loss consistently improves realism-oriented metrics (MPJPE, FID), (2) MoFA remains competitive under both settings without additional architectural changes, and (3) the changes in Diversity and Traj.err reflect a meaningful physical trade-off rather than a failure mode. This supports that our approach is compatible with contact-based constraints and benefits from them in a predictable and interpretable manner.

