# OpenReview forum: "MoFA: Dual-Task Motion Factorization for Human Motion Synthesis with Imperfect and Limited Data"
_ICLR.cc/2026/Conference — Submitted to ICLR 2026_

### Official Review · Reviewer_3qHX · 2025-10-29

**Soundness:** 2
**Presentation:** 2
**Contribution:** 2
**Rating:** 4
**Confidence:** 2

**Summary:**

In this paper, it proposed to utilize both high-quality and low-quality data for training. It decompose motion synthesis into local motion completion and trajectory adaption. The proposed method has been compared with many prevailing method and shows better performance than competitors.

**Strengths:**

1. The proposed method shows better performance than many prevailing methods.
2. It has conducted many ablation studies.

**Weaknesses:**

1. Local motion synthesis and global trajectory alignment has widely been utilized in different motion synthesis framework.
2. There is no explicit description of test dataset. It is hard to judge the performance without explicit explanation of test dataset.
3. Some figures are quite small. For instance, it is hard to read the Figure 4. In addition, Figure 1 is not illustrative, making it hard to understand the main contribution.
4. According to the visualization, ProMoGen has better performance than the proposed method. The improvement over ProMoGen is not obvious.

**Questions:**

NA

---

> ### Author Response · Authors · 2025-11-20
> **We thank the reviewers and provide additional clarification.**
>
> We sincerely thank the reviewer for your careful evaluation and constructive feedback, and we will make every effort to bridge any remaining gaps in understanding and address all raised questions directly and transparently.
>
> Except direct replies, we also have added supplementary experiments and analyses in $Appendix  Rebuttal$ to further clarify the contributions of our method. Specifically, the new section includes: (1) a brief preface outlining the purpose of the additional materials; (2) C.1 How does the inclusion of low-quality data improve overall performance? (3) C.2 How capable is the model itself? (4) C.3 Inference-related details of the model. In addition, we provide a new set of videos demonstrating the intentional mismatch between keyframes and trajectories, with discussion included in C.4-The performance when keyframes and trajectories are intentionally mismatched. We hope these supplemental results further support the validity of our approach and help reviewers better interpret the behavior and benefits of the our work.

---

> ### Author Response · Authors · 2025-11-20
> **Local motion synthesis and global trajectory alignment has widely been utilized in different motion synthesis framework.**
>
> Thank you for raising this important question. While the notions of motion hierarchy and local–global synthesis do appear in prior work, our contribution differs fundamentally in how the controlled motion generation problem itself is formulated.
>
> Most existing approaches introduce hierarchical structures in the feature space but still learn a single conditional distribution, implicitly requiring the network to resolve the interaction between keyframes and trajectories within one generative objective. In contrast, our method begins with a probabilistic reformulation of the task and explicitly separates the generative process into two sub-distributions with distinct responsibilities: one for keyframe-driven local motion completion and one for trajectory-driven global adaptation. This factorization introduces a structural separation of control signals rather than a feature-level hierarchy, enabling the model to remain coherent even when trajectory and keyframes are inconsistent at inference. Although the conceptual theme of “local to global” is shared with prior work, our approach is differentiated by treating local and global motion as two separate probabilistic components rather than a single entangled distribution.

---

> ### Author Response · Authors · 2025-11-20
> **Description of test dataset**
>
> Our work involves four datasets: HumanML3D and CombatMotion, which originate from high-fidelity motion capture or professionally curated game assets. Both datasets exhibit stable global trajectories and smooth joint motions, and are therefore treated as high-quality sources. In contrast, MotionX++ and an AIST++ subset are collected from vision-based mocap systems and suffer from noticeable motion smoothing artifacts and global trajectory jitter; hence, they are regarded as low-quality sources in our setting.
>
> To make effective use of these data, we adopt a probabilistic quality-labeling mechanism during data loading:
>
> 1) Samples from HumanML3D or CombatMotion are labeled as $Q_h$ with probability 0.8, and as $Q_l$ with probability 0.2. This design ensures that high-quality motion contributes to both global-trajectory-aware generation and local-motion learning, while still being predominantly used to strengthen trajectory-related components.
> 2) Samples fromMotionX++ or AIST++ are consistently labeled as $Q_l$, and are only used for local-motion learning due to their unreliable trajectory information.
>
>
> Importantly, regardless of the $Q_{h}$/$Q_l$ assignment above, each training session loads the entire dataset. We then perform a reproducible split using "sklearn.model_selection.train_test_split" with parameter "random_state = 42". This guarantees that evaluations are repeatable across runs and that conclusions are not reliant on a particular data split.

---

> ### Author Response · Authors · 2025-11-20
> **About the weakness 4.**
>
> The goal of our work is not to outperform existing models by a large margin, but to redefine the problem in a way that preserves competitive performance (slightly ahead of baselines, as shown in Appendix C.2.1) while simplifying the learning space. This reformulation enables the model to naturally accommodate low-quality data at the architectural level (Appendix C.1) and to remain robust when trajectory and keyframes are mismatched during inference (Appendix C.4).

---

> ### Author Response · Authors · 2025-11-27
> **Weakness 3. Quite small figures**
>
> We adjusted the layout of the oversmall image(Figure 4) and added a tenth page to explore the relationship between the model's functionality and foot contact loss.

---

> ### Comment · Reviewer_3qHX · 2025-11-28
>
> Thank you for your response. However, your rebuttal has not adequately addressed my concerns. Regarding the dataset, I am more concerned about the rationality of the test set and the experimental design, rather than the training set you listed. Your statement about "The goal of our work is not to outperform existing models by a large margin" is quite confusing to me. Furthermore, the differences between your method and previous approaches in terms of local motion synthesis and global trajectory alignment have not been clearly explained.

---

> > ### Author Response · Authors · 2025-11-28
> > **About test set.**
> >
> > We thank the reviewer‘s question for test set. We clarify our evaluation protocol as follows:
> >
> > 1. **The entire dataset is loaded once, including both high-quality and low-quality samples.**
> >    Each sample is stored as a dictionary structure containing the key `mark`, which indicates whether the motion annotation is $Q_h$ (*high-quality*, with complete supervision) or $Q_l$ (*low-quality*, with incomplete supervision).
> >
> > 2. **The train/test split is performed *before* training and remains fixed for all experiments.**
> >    We apply `sklearn.model_selection.train_test_split (random_state = 42)` with an 8:2 ratio on the **full dataset**, ensuring full reproducibility. At this stage, the complete test set - containing both high- and low-quality samples - is finalized and is never modified afterward. During testing, we consistently load the 100th-epoch checkpoint for all models to ensure a fair comparison. We chose the 100th-epoch checkpoint based on the observed training loss curves: although it may not be the point of absolute minimum loss for every model, it already reflects stable convergence for every method and provides a uniform evaluation reference across methods.
> >
> > 3. **Filtering low-quality samples in the test set for metric validity(by using key `mark`).**
> >    Quantitative evaluation requires complete ground-truth motion information. Since low-quality samples lack parts of the supervision signal, they cannot support needed numerical metrics. Therefore, **quantitative scores are computed only on the high-quality subset of the already-fixed test set**, without altering its composition.
> >
> > This evaluation protocol guarantees:
> >
> > * No distribution shift between training and evaluation
> > * No test data selection or filtering bias
> > * Full reproducibility of the reported results

---

> > ### Author Response · Authors · 2025-11-28
> > **About experimental design**
> >
> > We briefly summarize how our experiments are designed and what each part is intended to demonstrate.
> >
> > The experiments follow a simple principle: **each experiment verifies one specific claim of the paper**.
> >
> > * **MoFA vs state-of-the-art models**
> >   To show that our method improves precision, realism, and controllability under identical inputs (same trajectories and keyframes).
> >
> > * **MoFA vs our own base model (without QADT)**
> >   To show that the gain comes from the dual-task training strategy.
> >
> > * **Robustness tests under 1 / 3 / 5 / 7 / 10 keyframes**
> >   To verify that the model behaves stably across different levels of supervision — not only in a single “favorable” scenario.
> >
> > * **Feature-level analysis (PCA + embedding shift)**
> >   To explain *why* factorization works internally, not just to present better numbers.
> >
> > * **Ablations on CIM, LMRS, and TAMI**
> >   To confirm that the full factorization pipeline, rather than any single component, is responsible for the performance improvement.
> >
> > In the rebuttal, we further added:
> >
> > * **Different ratios of low-quality data**
> >   QADT consistently improves performance, confirming that the dual-task design — not data volume — is responsible for the benefits.
> >
> > * **Training without curriculum learning**
> >   Random keyframe sampling (1–20) still converges well, demonstrating that MoFA itself has enough ability.
> >
> > * **Fusion version vs MoFA**
> >   (keeping all modules but removing factorization) to isolate the effectiveness of the decomposition design.
> >
> > * **Effect of foot-contact loss**
> >   To show that MoFA remains physically realistic when such a constraint is added.
> >
> > **In summary:**
> > Our experiments are not designed to be complicated; rather, each is included to answer one clear question — *Does it outperform prior methods? Does it outperform our own base? Is it robust? Why does it work internally? And does it still work under different training and physical conditions?*

---

> > ### Author Response · Authors · 2025-11-28
> > **For the question "The goal of our work is not to outperform existing models by a large margin" is quite confusing to me."**
> >
> > The main goal of our work is **not to push absolute numerical scores as high as possible**, but to **redefine the controlled motion generation problem at the distribution level** — by explicitly disentangling local and global motion into two probabilistic sub-tasks. The motivation is that, under this formulation, the learning difficulty becomes significantly lower, and low-quality data (which cannot fully supervise motion) can still be used meaningfully.
> >
> > In other words, the central contribution of MoFA is **a problem-level reformulation and decoupled learning strategy**, not an attempt to “squeeze out” a few more points on standard benchmarks. The performance gains on the metrics are the *result* of this modeling principle, rather than the *goal* of the method.

---

> > ### Author Response · Authors · 2025-11-28
> > **the differences between your method and previous approaches in terms of local motion synthesis and global trajectory alignment**
> >
> > The contribution of our paper is **not** to add yet another *“local–global architecture”*. The key difference lies in **how the motion generation problem is formulated**. In existing methods, keyframes and trajectories are modeled within **one single conditional distribution**: $P(m|t,k)$.
> >
> > The network must implicitly resolve how local and global constraints interact, and when the two controls contradict at inference, they compete inside the same generative space with no explicit mechanism to maintain coherence.
> >
> > Our work begins with a **probabilistic reformulation of the task**: $P(m_2 \mid t, k) = \int P(m_2 \mid t, m_1) , P(m_1 \mid t, k) , dm_1$. with the conditional independence assumption: $m_2 \perp k \mid (t, m_1)$
> >
> > This changes the nature of the problem:
> >
> > * $(P(m_1 \mid t, k))$ captures **all keyframe-driven local variation**, and
> > * $(P(m_2 \mid t, m_1))$ enforces **trajectory-driven global consistency**.
> >
> > In addition, other works' "local-to-global" is either a hierarchical feature design or a coarse-to-fine architecture.
> > We are **decomposing the generative distribution itself**, and the two stages in MoFA directly implement the two distributions above — not a feature pyramid or a manually engineered hierarchy.
> >
> > This theoretical difference produces practical consequences:
> > when the trajectory and keyframes diverge at inference, prior approaches is hard to reconcile both signals inside one latent space, whereas MoFA has a built-in mechanism to maintain coherence.

---

### Official Review · Reviewer_UtVH · 2025-10-31

**Soundness:** 2
**Presentation:** 2
**Contribution:** 2
**Rating:** 4
**Confidence:** 4

**Summary:**

This paper proposes MoFA, a diffusion-based Motion Factorization framework for controllable human motion synthesis with potentially imperfect and limited data. The core idea is to decompose generation into two conditional sub-tasks:

Local Motion Completion (driven mainly by keyframes), and Trajectory Adaptation (aligning the local motion to a global path).

The system integrates a Local Motion Refinement Stack (LMRS) and Trajectory-Aware Motion Integration (TAMI), plus a Quality‑Aware Dual Training (QADT) scheme that leverages both high‑quality (reliable trajectories) and low‑quality (unreliable trajectories) data by replacing trajectories with a learned embedding when needed. Architecture and training flow are illustrated in Figure 1 (p. 3), losses in Eq. (12), and datasets/metrics in §4. Experiments show consistent improvements over recent baselines across multiple keyframe counts (K=1,3,5,7,10), with the strongest results reported at K=5 in Table 1 and additional extensive comparisons in Tables 4–8. The paper also provides qualitative examples (e.g., backflips, falls) in Figure 3, and a feature-level analysis supporting the factorization in Figure 5

**Strengths:**

The paper is well written with some nice figures. It addresses a practical problem about motion synthesis training with imperfect data.

**Weaknesses:**

Weaknesses

Novelty about imperfect data: The claim of training motion synthesis models with imperfect data captured from video has already been demonstrated in Actor (ICCV21) - could the authors please clarify how their strategy differs from Actor - in the abstract sense of using imperfect data for training.

Novelty vs prior “local-to-global” formulations
While the paper’s decomposition is well argued, it resembles earlier hierarchical or local-to-global pipelines (for example, local refinement followed by trajectory or scene alignment). The related work is cited, but the specific modeling advance over hierarchical diffusion or curriculum-based anchor methods is not fully disentangled. Clearer positioning about what MoFA can represent that prior pipelines cannot would strengthen the originality claim (see Section 2).

Independence assumption is strong and only qualitatively probed
The central claim is that the final global motion does not depend on keyframes once local motion is inferred. The support offered (Figure 5 with PCA and embedding statistics) is suggestive rather than conclusive. A quantitative test—such as measuring performance changes when keyframe features are also provided to the global stage via a residual cross-attention or gating—would better validate that keyframes add negligible signal after the local stage.

Fairness of comparisons when using more data
QADT uses additional low-quality data. It is unclear whether top baselines in Tables 1 and 4–7 were trained with the same expanded training set or at least the same number of sequences. Table 3 contrasts “Base” (high-quality only) with MoFA (with QADT), but this does not address whether external baselines benefited from the same data budget. This leaves ambiguity about how much of the gain comes from method design versus data scale.

Ambiguity or inconsistency in dimensionality notation
Section 3.3 says low-quality inputs have f1 features and high-quality inputs have f2 features, with high-quality exceeding low-quality by six channels. Appendix A.4, however, lists high-quality as 138 features and low-quality as 132 features. The two statements contradict each other and read like a notation swap or typo that should be corrected.

Choice of diffusion head and prediction target
Appendix A.2 states that the diffusion heads output motion representations instead of noise. This deviates from common DDPM practice and could affect stability or sampling. There is no ablation comparing noise prediction, velocity prediction, or direct data prediction to justify the chosen target.

Contact handling remains light
Although foot sliding and joint smoothness are reported (Tables 2–3), there is no explicit contact loss or contact consistency metric beyond foot sliding. Adding contact precision/recall or foot velocity thresholds during stance would strengthen claims about physical plausibility.

Efficiency and sampling details
The paper mentions a 1,000-step cosine schedule and some training details (Appendix A.2–A.3), but it omits inference sampling steps, wall-clock throughput, and parameter counts. Given practical importance, these should be included.

Mixed signals on Motion Fluency
In Table 2, Motion Fluency is sometimes lower than a baseline even though higher is claimed to be better. The text suggests average improvements, but this is not consistently visible across different numbers of keyframes. The definition of the metric and its correlation with perceived quality should be clarified.

Reproducibility gaps
Loss weights in the composite objective are not enumerated. FID computation details (feature backbone, whether the metrics network is pretrained or trained) and Diversity computation specifics are not fully described. Releasing code and an evaluation harness would meaningfully improve reproducibility.

**Questions:**

Validate the independence claim
Please add an ablation where keyframe features are also provided to the global module (for example, via a small residual cross-attention or gating). Report the change in MPJPE, FID, and trajectory error. If performance improves, it would suggest the final motion still depends on keyframes after the local stage. A complementary conditional-dependence probe (for example, a contrastive probing setup) would help quantify any remaining dependence.

Fair data budget comparison
Clarify whether external baselines in Table 1 and Tables 4–7 were retrained on exactly the same union of datasets used by MoFA with QADT. If not, please add results where a strong baseline (for example, StableMoFusion or PMG) is trained with the same additional low-quality data so the architectural gains can be separated from data-scale gains.

Definition and selection of “low-quality” data
How are sequences labeled as low-quality versus high-quality? Is there an automated thresholding scheme (for example, trajectory smoothness, jerk, missing markers) or manual curation? Please provide counts, sequence-length distributions, and an ablation showing performance as a function of the amount of low-quality data used.

Diffusion prediction target
You state that the generators output motion representations rather than noise. Are you training with a standard noise-prediction loss, a data-prediction loss, a velocity-style target, or a hybrid? Please specify and add an ablation comparing these choices and their impact on convergence and sample quality.

Notation consistency for feature dimensions
Please fix the mismatch between Section 3.3 and Appendix A.4 regarding feature dimensions and confirm the six additional trajectory channels. Even if minor, this inconsistency is confusing for reproduction.

Robustness under extreme mismatch
Beyond the qualitative examples in Figure 3, please include a stress test where keyframes and trajectories are deliberately inconsistent (for example, facing backward while the trajectory turns sharply). Quantify failure rates for foot sliding, trajectory deviation, and pose drift relative to baselines.

Contact metrics and constraints
Would adding a contact-aware loss (for example, low foot velocity during stance, ground-penetration penalties) further reduce foot sliding without hurting diversity? A small ablation would strengthen claims about physical realism.

Interpreting Motion Fluency
Please define Motion Fluency precisely and explain why higher values are desirable. In Table 2, MoFA’s Motion Fluency is sometimes lower than a baseline; does this correlate with human preference or perceived smoothness?

Compute and efficiency details
What are parameter counts, FLOPs, and inference latency (for example, number of sampling steps and seconds per four-second clip on the reported GPU)? Is there a fast sampler or distilled variant compatible with the two-stage design?

Curriculum and loss ablations
The progressive keyframe curriculum in Appendix A.5 appears helpful. Please add an ablation removing it, and another varying the loss weights, to quantify their influence.

Where QADT helps most
In Table 3, gains appear across different numbers of keyframes. Breaking down improvements by action class or motion type (for example, locomotion versus acrobatics) would show which behaviors benefit most from low-quality local supervision.

Failure cases and limitations
Please add a short discussion and figure of typical failure cases, such as self-intersections, drift on very long trajectories, or instability with very sparse keyframes. This would make the paper more balanced and actionable.

Overall recommendation (tentative)

Lean reject. The paper tackles a practical problem—entangled controls and imperfect data—with a clean factorization and a useful training strategy. The two most important items for rebuttal are: (1) a fair data-budget comparison where a strong baseline is trained on the same expanded data, and (2) a quantitative probe of the independence claim by allowing keyframes into the global stage.

I also think the paper is being mis-sold: the idea of using imperfect data for training has been shown to work before; the hierarchal decomposition of motion has also been shown before. I am not quite sure I see what this paper adds to the current literature.

---

> ### Author Response · Authors · 2025-11-20
> **We thank the reviewers and provide additional clarification.**
>
> We sincerely thank the reviewer for your careful evaluation and constructive feedback, and we will make every effort to bridge any remaining gaps in understanding and address all raised questions directly and transparently.
>
> Except direct replies, we also have added supplementary experiments and analyses in $Appendix  Rebuttal$ to further clarify the contributions of our method. Specifically, the new section includes: (1) a brief preface outlining the purpose of the additional materials; (2) C.1 How does the inclusion of low-quality data improve overall performance? (3) C.2 How capable is the model itself? (4) C.3 Inference-related details of the model. In addition, we provide a new set of videos demonstrating the intentional mismatch between keyframes and trajectories, with discussion included in C.4-The performance when keyframes and trajectories are intentionally mismatched. We hope these supplemental results further support the validity of our approach and help reviewers better interpret the behavior and benefits of the our work.

---

> ### Author Response · Authors · 2025-11-20
> **Validate the independence claim.**
>
> Thank you for the reviewer’s insightful comments. To further examine the conditional independence hypothesis, we added a controlled comparison in Appendix C.2.2 -- Verify the Validity of the Conditional Independence Hypothesis. Using the same evaluation dataset, we compare the original MoFA with a variant in which keyframe features are re-injected after the LMRS module for the subsequent global generation stage. The results show that the original MoFA achieves better reconstruction metrics, while the re-injection variant yields better FID. This indicates that MoFA behaves as a more conservative and reliability-oriented architecture: by enforcing a clean factorization and preventing the global stage from being influenced again by keyframe inputs, it minimizes accumulated prediction errors and produces more stable motion, even at the cost of slightly reduced output diversity. For human motion generation, such robustness is typically more desirable than uncontrolled creativity, further supporting the semantic and empirical soundness of the proposed decomposition.

---

> ### Author Response · Authors · 2025-11-20
> **The difference between our MoFA and ACTOR(ICCV21)**
>
> We read the paper you mentioned, which is "Action-Conditioned 3D Human Motion Synthesis with Transformer VAE". We think our QATD strategy differs from ACTOR. The two approaches differ fundamentally in their assumptions, objectives, and technical pathways: ACTOR performs single-fidelity denoising and augmentation, whereas our framework performs multi-fidelity joint modeling.
>
> ACTOR can be characterized as a VAE-based post-processing and data augmentation pipeline. It assumes that all training data originate from a single distribution, where noise is primarily random and isotropic. Under this assumption, VAE reconstruction provides a form of "statistical averaging," smoothing high-frequency noise. However, this procedure cannot remove systematic errors-most notably in the global trajectory-which persist after reconstruction. When such partially denoised data are further used for large-scale augmentation, these residual biases introduce distributional drift. As acknowledged by ACTOR's authors, the denoised sequences produced by ACTOR remain distributionally misaligned with clean ground-truth motion.
>
> In contrast, our QATD follows a multi-fidelity joint generative modeling paradigm. Rather than attempting to “clean” noisy data, we explicitly leverage the structural properties of human motion and the heterogeneous reliability of different data sources. Specifically, we construct a generative model with an explicit local/global decomposition, where low-fidelity data only supervise the reliable local motion subspace, while the global trajectory is trained exclusively from high-fidelity data. This design prevents systematic low-fidelity data's trajectory noise from propagating into the generative model and yields a learning process that is more consistent, interpretable, and stable.

---

> ### Author Response · Authors · 2025-11-20
> **Fair data comparison**
>
> Thank you for the constructive suggestion. The comparative results under fair and unified data conditions are presented in Appendix C.2.1 -- Fair Comparison with Other Models. The experiments demonstrate that even when all models are trained by using identical data, our approach still achieves superior reconstruction quality compared to the competing methods.

---

> ### Author Response · Authors · 2025-11-20
> **Definition and selection of “low-quality”  data.**
>
> Our work involves four datasets: HumanML3D and CombatMotion, which originate from high-fidelity motion capture or professionally curated game assets. Both datasets exhibit stable global trajectories and smooth joint motions, and are therefore treated as high-quality sources. In contrast, MotionX++ and an AIST++ subset are collected from vision-based mocap systems and suffer from noticeable motion smoothing artifacts and global trajectory jitter; hence, they are regarded as low-quality sources in our setting.
>
> To make effective use of these data, we adopt a probabilistic quality-labeling mechanism during data loading:
>
> 1) Samples from HumanML3D or CombatMotion are labeled as $Q_h$ with probability 0.8, and as $Q_l$ with probability 0.2. This design ensures that high-quality motion contributes to both global-trajectory-aware generation and local-motion learning, while still being predominantly used to strengthen trajectory-related components.
> 2) Samples fromMotionX++ or AIST++ are consistently labeled as $Q_l$, and are only used for local-motion learning due to their unreliable trajectory information.
>
>
> Importantly, regardless of the $Q_{h}$/$Q_l$ assignment above, each training session loads the entire dataset. We then perform a reproducible split using "sklearn.model_selection.train_test_split" with parameter "random_state = 42". This guarantees that evaluations are repeatable across runs and that conclusions are not reliant on a particular data split.

---

> ### Author Response · Authors · 2025-11-20
> **Diffusion prediction target.**
>
> We would like to clarify that our work follows the standard practice in human motion diffusion models, where the model predicts clean motion rather than noise. This convention has been widely adopted in related literature because clean-motion prediction allows the learning signal to be expressed in physically meaningful terms (e.g.,  joint constrains, and trajectory constrains), whereas pure noise prediction relies almost exclusively on an MSE constraint in the noise space and cannot incorporate such semantics.
>
> We acknowledge that a systematic investigation of training objectives (noise vs. clean data vs. velocity vs. hybrid) is scientifically valuable. However, this topic is orthogonal to the main contribution of our paper and would require substantial additional experimentation and design—not merely a toggle in loss functions—because it fundamentally changes the model’s supervision space and the set of feasible structural constraints, in addition, it also needs a large amount of potential Parameter Tuning works. We believe this direction merits a stand-alone, dedicated empirical study rather than an auxiliary comparison inside the present paper. For this reason, and given the limited rebuttal period, we refrain from adding such an extensive experimental branch here.

---

> ### Author Response · Authors · 2025-11-20
> **fix the mismatch between Section 3.3 and Appendix A.4**
>
> The issue has now been resolved.

---

> ### Author Response · Authors · 2025-11-20
> **Stress test and failure case**
>
> We have updated the supplementary material with new videos illustrating the model’s behavior under mismatched control conditions. The accompanying explanations and interpretation of these visualizations are provided in Appendix C.4 - Explanation of the Visual Video.

---

> ### Author Response · Authors · 2025-11-20
> **Contact metrics and constraints.**
>
> Thank you for the thoughtful comment. We fully agree that incorporating additional physical constraints--such as contact-aware loss--can further improve motion quality. Prior works benefit from the complete HumanML3D format, which provides rich supervision (e.g., foot-ground contact labels, linear and angular velocities), enabling strong kinematic priors that naturally suppress foot sliding. In contrast, our setting is designed to fuse on low-quality vision-based motion datasets, where such contact information is unavailable and global velocity signals are highly noisy. Therefore, we adopt a lightweight representation based solely on joint rotations and trajectories to ensure broad applicability across imperfect datasets. While heuristic reconstruction of Humanml3D-style contact labels from foot velocities is theoretically possible, it would require extensive preprocessing and optimization beyond the intended scope of this work. We view contact integration as an engineering enhancement rather than a conceptual or method issues, in addition, our architecture can readily incorporate such constraints in future work without redesigning the overall framework.

---

> ### Author Response · Authors · 2025-11-20
> **About Motion Fluency.**
>
> Motion Fluency  is defined as the average autocorrelation over a short temporal lag, where higher values indicate smoother and more temporally coherent motion. Although MoFA shows slightly lower values than the baseline in a few entries, the difference does not cause noticeable perceptual degradation, since human preference depends on more aspects than temporal smoothness alone.

---

> ### Author Response · Authors · 2025-11-20
> **Compute and efficiency details**
>
> Thank you for the thoughtful question. The Appendix C.3 - Model Inference Related Information includes comparative tables reporting inference speed, model complexity, sampling steps and latency. We hope these sections clearly address the your concerns.

---

> ### Author Response · Authors · 2025-11-20
> **Remove Curriculum and loss ablations.**
>
> Thank you for the reviewer’s suggestion. A fair comparison with other models is provided in Appendix C.2.1. We also claim that, throughout the newly added Appendix C, curriculum learning has been fully removed; instead, we adopt random sampling of the number of keyframes during data loading to improve generalization while ensuring a consistent and unbiased comparison protocol across all methods.
>
> We view loss-weight tuning as an implementation detail rather than a core contribution of the paper. For completeness, we clarify our choices: in the initial submission, the weights for rot, traj, and joint losses were set to 1.0, while the acc and vel losses were set to 0.5. Subsequent experiments showed that these weights bring overly strong local consistency, which inadvertently increased foot-sliding artifacts. Our current configuration decreases the acc/vel losses' weights to 0.1 and applies a 0.8 scaling factor to the local-motion loss when low-quality data participates in gradient updates, encouraging the model to consider global motion more strongly. This adjustment consistently improves motion quality, but we emphasize that it does not affect the conceptual contribution of the work.

---

> ### Author Response · Authors · 2025-11-20
> **Which behaviors benefit most from low-quality local supervision.**
>
> From a data perspective, large-amplitude and highly exaggerated poses represent a substantially more difficult generation regime, and models benefit most from exposure to such challenging motions. Due to hardware and participant limitations, physical/optical MoCap datasets are dominated by everyday movements and contain very few high-difficulty actions. In contrast, video-based MoCap enables access to a vast amount of dynamic and complex motions. Incorporating these low-quality yet diverse sequences strengthens the model’s ability to perceive and reproduce high-difficulty actions, ultimately improving its expressiveness and coverage.

---

> ### Author Response · Authors · 2025-11-26
> **The introduction of contact metrics and constraints**
>
> Thank you for your interest and positive feedback. Below, we provide two sets of small-scale experiments try o directly address the issues you raised.
>
> $\textbf{E1: The smoothness, fluency and physically foot-sliding results}$
> | Method | FS(cm) ↓ | JS ↓ | MF → | Method | FS(cm) ↓ | JS ↓ | MF → |
> |--------|---------|------|------|--------|---------|------|------|
> | **MDM-5** | **3.815** | **0.006** | **0.916** | **MDM-10** | **3.853** | **0.006** | **0.914** |
> | StableMoFusion-5 | 3.844 | 0.013 | 0.785 | StableMoFusion-10 | 3.862 | 0.013 | 0.781 |
> | PMG-5 | 3.816 | 0.009 | 0.841 | PMG-10 | 3.893 | 0.011 | 0.830 |
> | MoFA-5 | 3.947 | 0.011 | 0.852 | MoFA-10 | 3.961 | 0.014 | 0.820 |
>
> The table reports Foot Sliding (FS), Joint Smoothness (JS), and Motion Fluency (MF) for four models (MDM, StableMoFusion, PMG, MoFA) under two guidance settings (5 keyframes and 10 keyframes), $\textbf{without using contact loss. }$Across all metrics and both keyframe conditions, MDM achieves the best scores. However, this does not imply that MDM is inherently superior. Rather, MDM tends to generate more conservative motions with smaller amplitude, which naturally results in lower accumulated foot sliding and higher smoothness/fluency. When interpreted together with FID and MPJPE results in the appendix, MDM does not outperform other methods in overall motion realism or precision.
>
> The differences among the four models in FS, JS, and MF are relatively small, indicating that our MoFA will not produce noticeable or systematic foot-sliding artifacts. These metrics therefore help confirm that the proposed framework does not unintentionally bias the generation toward unstable contact or discontinuous motion.
>
> $\textbf{E2: What happened if we introduce foot-contact loss?}$
> | Method | FS(cm) ↓ | JS ↓ | MF → | Method | FS(cm) ↓ | JS ↓ | MF → |
> |--------|---------|------|------|--------|---------|------|------|
> | **MDM-5** | **0.557** | **0.003** | **0.890** | MDM-10 | 0.565 | **0.003** | **0.895** |
> | MoFA-5 | 0.558 | 0.004 | 0.781 | **MoFA-10** | **0.562** | 0.005 | 0.747 |
>
>
> The table reports Foot Sliding (FS), Joint Smoothness (JS) and Motion Fluency (MF) for MDM and MoFA when contact loss is applied (using the same formulation as in MDM). We evaluate both models under 5-keyframe and 10-keyframe guidance.
>
> Compared with the table without contact loss, FS drops substantially for both models, indicating that contact loss is effective in reducing foot–ground drifting. Importantly, our model adapts to this constraint without architectural modification and remains close to MDM in all three metrics across both guidance settings. MDM still scores slightly better overall, which is consistent with the explanation provided earlier: MDM tends to generate more conservative, low-amplitude motions, naturally resulting in lower accumulated FS and higher smoothness/fluency.
>
> We also observe that adding contact loss generally improves JS (less abrupt acceleration) while slightly decreasing MF (reduced long-range temporal continuity). This trade-off is intuitive: adding a physical constraint stabilizes ground contact and suppresses sudden movements, but also mildly restricts the temporal variability of motion. Crucially, the differences remain small, and both models produce stable contact and continuous motion, demonstrating that the proposed method does not introduce foot-sliding artifacts and is compatible with contact-based physical regularization.

---

### Official Review · Reviewer_R7HS · 2025-10-31

**Soundness:** 3
**Presentation:** 3
**Contribution:** 3
**Rating:** 6
**Confidence:** 4

**Summary:**

This paper proposes MoFA, a diffusion-based Motion Factorization framework for controllable human motion synthesis when trajectory and keyframe controls are imperfect or unpaired. The core idea is to decompose generation into (i) Local Motion Completion from keyframes (and optionally trajectory) using a Local Motion Refinement Stack (LMRS), and (ii) Trajectory Adaptation that aligns the local motion to a path via Trajectory-Aware Motion Integration (TAMI). A Quality-Aware Dual Training (QADT) regime exploits high-quality datasets for both stages and low-quality datasets (with unreliable trajectories) for the local stage via a learnable “trajectory” embedding. On HumanML3D/CombatMotion (global) plus AIST++/MotionX++ subset (local), MoFA reports consistent gains over strong baselines across MPJPE, FID, Trajectory Error, K-MPJPE, foot sliding, and smoothness for K∈{1,3,5,7,10} keyframes, with detailed architecture, loss, and training schedules provided.

**Strengths:**

- Clean formulation that disentangles keyframe-driven local completion from global trajectory adaptation; QADT is a pragmatic way to mine low-quality motion data without harming the global branch.
- Consistent quantitative gains across MPJPE/FID/Traj.err/K-MPJPE under multiple K, plus perceptual metrics (foot sliding, smoothness, autocorrelation). Comparisons include recent diffusion baselines and a controlled “Base” variant.
- Architecture blocks (KPE/GTE, CIM, LMRS, TAMI, LMG/GMG), losses (L_{\text{rot}},L_{\text{traj}},L_{\text{vel}},L_{\text{acc}},L_{\text{joint}}), data representation (22 joints × 6D + 6 trajectory dims), and training schedule are explicitly stated.
- Addresses a real deployment pain-point—keyframe/trajectory entanglement and data quality scarcity—showing improved stability without sacrificing diversity.

**Weaknesses:**

- Evidence for “unpaired control” robustness is limited. While the motivation centers on mismatched keyframes/trajectories at inference, experiments mainly use paired ground truth (Qh) for evaluation; explicit mismatch tests (e.g., swap trajectories between clips) are missing.
- QADT relies on a binary Qh/Ql partition but criteria and label noise sensitivity are under-specified. An ablation varying the fraction/corruption of Ql or mislabels would strengthen the claim.
- Several compared methods are not designed for joint keyframe+trajectory control; adding targeted control baselines (e.g., TLControl or adapting OmniControl with keyframes) would better isolate MoFA’s benefits. Ensure identical keyframe/trajectory protocols and report wall-clock/sample counts.
- FID for motion features, Diversity, JS, and foot-sliding need precise definitions (feature extractor, windowing, units). Report exact training/seed counts and CIs; some tables show small SDs without detailing runs.
- Beyond Base vs. MoFA, a finer ablation (LMRS only / TAMI only / QADT off) and swapping the 6-channel trajectory encoding would clarify which parts drive which metrics.

**Questions:**

1. Can you report results when keyframes and trajectories are intentionally mismatched (e.g., random trajectory with fixed keyframes)? Does MoFA degrade gracefully vs. direct-fusion baselines?
2. How are Qh/Ql labels assigned in practice? What happens if 10–30% of Ql actually contain reliable trajectories (or vice versa)? Please add sensitivity curves.
3. What exactly are the 6 trajectory channels (XY position + facing + height + velocities?) and at what rate? Any benefit to spline or velocity-only encodings?
4. Results for (LMRS only), (TAMI only), and (QADT off) under K={1,5,10} would help attribute gains.
5. Training/inference speed vs. baselines, guidance cost for larger K, and sampling steps? Any trade-offs with DiT depth/heads?
6. Do results transfer to in-the-wild or longer sequences beyond benchmark stats? Any failure modes (e.g., sharp turns, jumping with sparse keyframes)?

---

> ### Author Response · Authors · 2025-11-20
> **We thank the reviewers and provide additional clarification.**
>
> We sincerely thank the reviewer for your careful evaluation and constructive feedback, and we will make every effort to bridge any remaining gaps in understanding and address all raised questions directly and transparently.
>
> Except direct replies, we also have added supplementary experiments and analyses in $Appendix  Rebuttal$ to further clarify the contributions of our method. Specifically, the new section includes: (1) a brief preface outlining the purpose of the additional materials; (2) C.1 How does the inclusion of low-quality data improve overall performance? (3) C.2 How capable is the model itself? (4) C.3 Inference-related details of the model. In addition, we provide a new set of videos demonstrating the intentional mismatch between keyframes and trajectories, with discussion included in C.4-The performance when keyframes and trajectories are intentionally mismatched. We hope these supplemental results further support the validity of our approach and help reviewers better interpret the behavior and benefits of the our work.

---

> ### Author Response · Authors · 2025-11-20
> **Q：Mismatched combination of keyframes and trajectories.**
>
> We have updated the supplementary material with new videos illustrating the model’s behavior under mismatched control conditions. The accompanying explanations and interpretation of these visualizations are provided in Appendix C.4 - Explanation of the Visual Video.

---

> ### Author Response · Authors · 2025-11-20
> **The influence of the Qh/Ql data.**
>
> We would first like to clarify that the low-quality data used in our work do not contain usable trajectory signals; since these sequences are from vision-based motion capture datasets, their global trajectories are heavily corrupted by drift, jitter, and missing segments, making them unsuitable to be directly incorporated as trajectory supervision during training. Regarding the influence of low-quality data $Q_l$, our experiments show a consistent trend: adding a small amount of $Q_l$ initially degrades performance, but as the proportion of $Q_l$ increases, the additional motion diversity progressively benefits learning and eventually leads to performance that surpasses the model trained solely on high-quality data. This suggests that large-scale low-quality data can provide complementary signals rather than noise when used within our framework.
>
> The detailed experiments results and figures could be seen in Appendix C.1-HOW DOES THE INCLUSION OF LOW-QUALITY DATA IMPROVE THE OVERALL PERFORMANCE OF THE METHOD.

---

> ### Author Response · Authors · 2025-11-20
> **Regarding the comparison model**
>
> We note that OmniControl is already included in our comparison. Although TLControl is related, its formulation is designed for "text + partial trajectory" joint control rather than the "trajectory + keyframe" setting targeted in our work. More broadly, most controllable motion generation methods rely on text as one of the modalities; however, text provides higher abstract semantic guidance and would confound the evaluation of fine-grained controllability. To ensure a fair and focused comparison, all baseline models are adapted by removing their text encoders and replacing them with the same trajectory and keyframe encoders as in MoFA, so that the fused representations enter the subsequent architectures under a consistent experimental protocol. This setup allows us to rigorously and fairly assess different models.

---

> ### Author Response · Authors · 2025-11-20
> **More Ablations for LMRS/TAMI/QADT off?**
>
> Thank you for the helpful suggestion. The component-wise ablation results of our model are already provided in Appendix Table 8.
>
> In addition, same to the Answer "The influence of the Qh/Ql data", the comparison and analysis related to QATD are discussed in Appendix C.1. We hope these supplementary results could offer a clear and complete illustrations.

---

> ### Author Response · Authors · 2025-11-20
> **Training/inference speed vs. baselines, guidance cost for larger K, and sampling steps? Any trade-offs with DiT depth/heads?**
>
> Thank you for the thoughtful question. The detailed descriptions of all architectural components are provided in Appendix A.2 — Network Architecture. In addition, Appendix C.3 — Model Inference Related Information includes comparative tables reporting inference speed, model complexity, sampling steps and latency. We hope these sections clearly address the your concerns.

---

> ### Author Response · Authors · 2025-11-20
> **Model generalization ability**
>
> Thank you for the reviewer’s thoughtful comments. Our current work does not yet support variable-length motion inputs; all models are trained on sequences of a unified length. Extending the framework to longer or autoregressive motion generation is an important direction that we plan to explore in future work.
>
> Regarding extreme cases, we include a representative failure example in the uploaded videos: a highly dynamic street-dance motion containing rapid rotations and lying outside the training distribution, where MoFA produces unrealistic turning and ground sliding. This illustrates a current limitation of the model and suggests that incorporating more domain-specific datasets (e.g., dance motions) may further improve robustness.

---

> ### Author Response · Authors · 2025-11-20
> **About the trajectory.**
>
> In our design, the trajectory representation does not encode velocity, acceleration, or other derived motion attributes. Instead, each trajectory frame is formatted as ([x, y, z, 0, 0, 0]), where the first three components denote the Cartesian position of the root and the remaining three zeros are introduced solely for dimensional alignment with the 6-D joint rotation representation. In other words, we intentionally restrict the trajectory channel to pure positional information in order to isolate and examine the influence of the trajectory signal itself on model training, without injecting additional motion cues such as speed or curvature. Although spline-based representations or velocity encoding could potentially provide smoother temporal signals or richer dynamics, they would implicitly introduce extra inductive bias and confound the interpretation of the trajectory’s individual contribution.

---

> ### Author Response · Authors · 2025-11-26
> **Further explanation of FID, JS, MF, and Foot sliding indicators**
>
> $\textbf{For FID}.$
> We follow the standard Fréchet distance protocol. For each real and generated motion sequence, we extract a 256-dimensional embedding using our pretrained motion encoder, whose parameters remain frozen during evaluation. We then compute the empirical mean and covariance of real and generated features separately and report the Fréchet distance values.
>
> The motion encoder is an 8-layer Transformer trained in a self-supervised contrastive manner on the motion dataset. Two stochastic augmentations (temporal masking and Gaussian noise) of the same motion form a positive pair, while other samples in the batch serve as negatives. Training follows an InfoNCE-style objective, encouraging invariance between augmented views of the same motion. After training, the encoder is used exclusively for feature extraction when computing FID.
>
> $\textbf{For JS (joint-smoothness).}$
> The JS metric evaluates the instantaneous physical smoothness of generated motion by computing the root-mean-square(RMS) joint acceleration. For each motion, we first convert 6D rotations to 3D joint positions, then compute joint velocities and accelerations via first- and second-order temporal differences.  Lower JS values correspond to smaller, less abrupt changes in velocity, indicating smoother and more physically stable motion.
>
> $\textbf{For MF (motion-fluency).}$
> The MF metric evaluates temporal continuity by measuring the average autocorrelation of joint trajectories. After converting each sequence to joint positions and flattening them spatially, we compute the autocorrelation coefficient between the trajectory signal and a time-shifted version (lag = 5 frames). This coefficient is averaged across joints and samples. Higher MF values indicate more consistent temporal evolution and fewer unnecessary direction changes, reflecting more fluent and natural motion.
>
> $\textbf{Importantly.}$
> JS and MF capture complementary aspects of motion dynamics: JS measures local smoothness by penalizing sudden accelerations, whereas MF measures global continuity by encouraging consistent temporal progression. They are not intended as absolute indicators of motion quality; a model could score well by generating overly conservative, low-amplitude motion. Their role is to quantify specific facets of physical plausibility, and they are interpreted in conjunction with other metrics such as FID and MPJPE.
>
> $\textbf{For Foot-sliding}$
> The foot-sliding metric quantifies how much the feet undesirably move across the ground during expected contact phases. For each sequence, we accumulate horizontal foot displacement only when a foot is close to the ground (i.e., below a height threshold), and ignore displacement when the foot is lifted. The accumulated movement is averaged across frames and feet and converted to centimeters. Lower values indicate more stable foot–ground contact and fewer physically implausible sliding artifacts.
>
> In practice, JS and MF show very small variance (typically <1e-3), as most generated motions are already highly smooth in terms of acceleration and temporal continuity. Foot-sliding, however, naturally exhibits much larger variance, because motions of different amplitude accumulate contact-phase displacement to very different degrees (small-range motions slide less, large-range motions slide more). Therefore, we mainly report the mean foot-sliding score, as its variance reflects motion type rather than model quality.

---

### Official Review · Reviewer_7uQa · 2025-11-01

**Soundness:** 3
**Presentation:** 2
**Contribution:** 2
**Rating:** 4
**Confidence:** 4

**Summary:**

The paper presents a motion factorization framework, MoFA, that decomposes the motion synthesis task into two complementary sub-tasks: (1) Local Motion Completion and (2) Trajectory Adaptation. MoFA comprises Local Motion Refinement Stack (LMRS) and Trajectory-Aware Motion Integration (TAMI) to perform local pose refinement and trajectory alignment. Furthermore, the paper introduces the Quality-Aware Dual Training (QADT) strategy, which utilizes low-quality data as auxiliary supervision, improving generalization. Experimental results show that MoFA can outperform previous methods in terms of frame-wise pose quality and trajectory alignment.

**Strengths:**

1. Although there is room for improvement in presentation, the paper itself is written well enough to make readers understand the authors' motivation, the proposed method, and the experimental results.
2. The proposed framework is well-designed for motion synthesis conditioned on keyframe pose and trajectory.
3. The experimental results demonstrate that the proposed method outperforms previous methods in terms of frame-wise pose quality and trajectory alignment.

**Weaknesses:**

1. When I saw the generated samples (supplementary material), I felt that the foot sliding was large. In my impression, the foot sliding was not so large in OmniControl or MotionLab. The proposed method should be compared to the previous methods also in terms of Foot Sliding.
2. L.198 (Section 3.2) says "the multimodal distribution $P(m_2|t, k)$ is simplified into two conditional distributions", but this is not obvious. An objective explanation of why the combination of two decomposed conditional distributions is simpler than the original distribution would be required. In my opinion, this decomposition is one of the reasons why the foot sliding is large, even though the decomposition facilitates frame-wise pose quality and trajectory alignment. So, it is not obvious that this decomposition is effective.
3. This is a minor weakness. Figures 1, 2, and 4 are a little fine. The figures are a little hard to see.

**Questions:**

I would appreciate it if the authors could address Weaknesses 1 and 2 that I provided above.

---

> ### Comment · Reviewer_7uQa · 2025-11-01
> **Minor comment**
>
> Minor issues with presentation
> - Throughout the whole document: **\citep{}** should be used instead of **\cite{}** or **\citet{}**.
> - L.145: "Diffusion Model(DM)" should be "Diffusion Model (DM)" (with a whitespace between "Model" and "(DM)").
> - Equation (11): "$LMRS(CIM(z_k, z_t))$" should be "$\text{LMRS}(\text{CIM}(z_k, z_t))$" (non-Italic).
> - L.256: It should be explained that $b$ is batch size and $l$ is sequence length.

---

> > ### Author Response · Authors · 2025-11-20
> > **Details have been optimized.**
> >
> > We have corrected the relevant errors. Thank you for pointing them out.

---

> ### Author Response · Authors · 2025-11-20
> **We thank the reviewers and provide additional clarification.**
>
> We sincerely thank the reviewer for your careful evaluation and constructive feedback, and we will make every effort to bridge any remaining gaps in understanding and address all raised questions directly and transparently.
>
> Except direct replies, we also have added supplementary experiments and analyses in $Appendix  Rebuttal$ to further clarify the contributions of our method. Specifically, the new section includes: (1) a brief preface outlining the purpose of the additional materials; (2) C.1 How does the inclusion of low-quality data improve overall performance? (3) C.2 How capable is the model itself? (4) C.3 Inference-related details of the model. In addition, we provide a new set of videos demonstrating the intentional mismatch between keyframes and trajectories, with discussion included in C.4-The performance when keyframes and trajectories are intentionally mismatched. We hope these supplemental results further support the validity of our approach and help reviewers better interpret the behavior and benefits of the our work.

---

> > ### Author Response · Authors · 2025-11-20
> > **Q：About the Foot Sliding.**
> >
> > We thank the reviewer for pointing out the foot-sliding issue. We fully agree that reducing foot sliding is a relevant goal in human motion generation. We would like to clarify that the sliding observed in our preliminary results has its own reasons and it could be alleviated.
> >
> > (1) Prior works such as OmniControl and MotionLab rely on the full HumanML3D format, which provides rich supervisory signals including foot–ground contact labels, velocities, and angular velocities. These strong kinematic priors effectively regularize foot planting and naturally reduce sliding. In contrast, our setup intentionally targets broader applicability across low-quality vision-based motion datasets, where foot-contact information is absent and global/rotational velocities are significantly noisier. For this reason we have to adopt a lightweight representation only based on joint rotation and trajectory for data alignment. Consequently, our results have more obvious sliding.
> >
> > (2) The combination of a smoothness regularizers and local motion learning in our method encourages temporal consistency but cannot fully eliminate foot sliding without explicit contact supervision.
> >
> > It is important to note that we think this is not a limitation of our model architecture, but rather of the available supervision. In fact, we verified that reducing the smoothness loss weight and lowering the importance of local-motion updates allows the model to focus more on high-quality trajectories and visibly alleviates sliding (shown in the new videos). Moreover, HumanML3D-style contact constraints can theoretically be reconstructed for low-quality datasets using heuristic thresholding based on foot velocity, but this requires extensive preprocessing and is beyond the intended scope of the current work.
> >
> > Overall, while foot sliding is a known challenge in motion generation and also appears in our task setting, it does not undermine the objective of this paper. In addition, the proposed method can also naturally and seamlessly incorporate explicit foot constraints in future extensions.

---

> ### Author Response · Authors · 2025-11-20
> **Q: An objective explanation of why the combination of two decomposed conditional distributions is simpler than the original distribution would be required.**
>
> Below we try to explain why learning the joint distribution $P(m \mid t,k)$ is "harder", and why the proposed factorization alleviates this difficulty.
>
> 1. Joint learning enforces two competing objectives simultaneously.
> A one-stage model that directly learns $P(m \mid t,k)$ must satisfy both (i) local keyframe pose recovery and (ii) global trajectory alignment in every denoising step. In real data, keyframes and trajectories are not always fully consistent, so these two supervision signals can produce conflicting gradients. The model is thus forced to solve two objectives that may pull the output in opposite directions.
>
> 2. Factorization removes direct competition between the two control signals. Each stage therefore focuses on a single coherent target, instead of solving two potentially contradictory targets at once.
>
> 3. Reduced functional complexity. The two conditional distributions primarily model keyframe-related and trajectory-related variations respectively. This reduces the complexity of the function each stage must approximate, which in turn eases optimization and stabilizes convergence.
>
> 4. According to the supplementary experiment,  C.2.2-VERIFY THE VALIDITY OF THE CONDITIONAL INDEPENDENCE HYPOTHESIS,  could further demonstrate this factorization 's effectiveness.
>
> In summary, our claim that the factorization is “simpler” refers to optimization behavior: decoupling two potentially conflicting objectives into two non-competing conditional distributions reduces gradient interference and makes each learning task more stable.

---

> ### Comment · Reviewer_7uQa · 2025-11-24
>
> Thank you for your work and replies. I went through all the reviewers' comments and your responses.
>
> Thanks to your response, I now understand the point of this paper deeply than before. The proposed framework enables us to effectively utilize two types of datasets: (1) high-quality data and (2) low-quality data. I acknowledge that this framework yields great frame-wise pose quality (as measured by K-MPJPE) and trajectory alignment (as measured by Trajectory Error).
>
> On the other hand, the authors and I may differ in how seriously we should take the foot-sliding issue. Currently, I suspect that MoFA achieves good K-MPJPE and Trajectory Error scores at the expense of Foot Sliding. This is why I asked to compare MoFA to the previous methods, also in terms of Foot Sliding. I am also interested in whether a foot contact loss (such as Equation 4 of the [MDM paper](https://openreview.net/forum?id=SJ1kSyO2jwu) or Equation 4 of the [StableMoFusion paper](https://dl.acm.org/doi/abs/10.1145/3664647.3681657)) can be effectively incorporated into MoFA. The fact that some papers (such as MDM, StableMoFusion, etc.) highlight such foot contact losses indicates that not a few people in this community pay a lot of attention to this issue.
>
> In addition to the above discussion, I would like to know what motion feature is used to compute FID scores. Reviewer R7HS also points out this issue.

---

> > ### Author Response · Authors · 2025-11-26
> > **Further explanation of FID, JS, MF, and Foot sliding indicators**
> >
> > $\textbf{For FID}.$
> > We follow the standard Fréchet distance protocol. For each real and generated motion sequence, we extract a 256-dimensional embedding using our pretrained motion encoder, whose parameters remain frozen during evaluation. We then compute the empirical mean and covariance of real and generated features separately and report the Fréchet distance values.
> >
> > The motion encoder is an 8-layer Transformer trained in a self-supervised contrastive manner on the motion dataset. Two stochastic augmentations (temporal masking and Gaussian noise) of the same motion form a positive pair, while other samples in the batch serve as negatives. Training follows an InfoNCE-style objective, encouraging invariance between augmented views of the same motion. After training, the encoder is used exclusively for feature extraction when computing FID.
> >
> > $\textbf{For JS (joint-smoothness).}$
> > The JS metric evaluates the instantaneous physical smoothness of generated motion by computing the root-mean-square(RMS) joint acceleration. For each motion, we first convert 6D rotations to 3D joint positions, then compute joint velocities and accelerations via first- and second-order temporal differences.  Lower JS values correspond to smaller, less abrupt changes in velocity, indicating smoother and more physically stable motion.
> >
> > $\textbf{For MF (motion-fluency).}$
> > The MF metric evaluates temporal continuity by measuring the average autocorrelation of joint trajectories. After converting each sequence to joint positions and flattening them spatially, we compute the autocorrelation coefficient between the trajectory signal and a time-shifted version (lag = 5 frames). This coefficient is averaged across joints and samples. Higher MF values indicate more consistent temporal evolution and fewer unnecessary direction changes, reflecting more fluent and natural motion.
> >
> > $\textbf{Importantly.}$
> > JS and MF capture complementary aspects of motion dynamics: JS measures local smoothness by penalizing sudden accelerations, whereas MF measures global continuity by encouraging consistent temporal progression. They are not intended as absolute indicators of motion quality; a model could score well by generating overly conservative, low-amplitude motion. Their role is to quantify specific facets of physical plausibility, and they are interpreted in conjunction with other metrics such as FID and MPJPE.
> >
> > $\textbf{For Foot-sliding}$
> > The foot-sliding metric quantifies how much the feet undesirably move across the ground during expected contact phases. For each sequence, we accumulate horizontal foot displacement only when a foot is close to the ground (i.e., below a height threshold), and ignore displacement when the foot is lifted. The accumulated movement is averaged across frames and feet and converted to centimeters. Lower values indicate more stable foot–ground contact and fewer physically implausible sliding artifacts.
> >
> > In practice, JS and MF show very small variance (typically <1e-3), as most generated motions are already highly smooth in terms of acceleration and temporal continuity. Foot-sliding, however, naturally exhibits much larger variance, because motions of different amplitude accumulate contact-phase displacement to very different degrees (small-range motions slide less, large-range motions slide more). Therefore, we mainly report the mean foot-sliding score, as its variance reflects motion type rather than model quality.

---

> > ### Author Response · Authors · 2025-11-26
> > **Foot-sliding comparison and the introduction of the foot contact loss**
> >
> > Thank you for your interest and positive feedback. Below, we provide two sets of small-scale experiments try o directly address the issues you raised.
> >
> > $\textbf{E1: The smoothness, fluency and physically foot-sliding results}$
> > | Method | FS(cm) ↓ | JS ↓ | MF → | Method | FS(cm) ↓ | JS ↓ | MF → |
> > |--------|---------|------|------|--------|---------|------|------|
> > | **MDM-5** | **3.815** | **0.006** | **0.916** | **MDM-10** | **3.853** | **0.006** | **0.914** |
> > | StableMoFusion-5 | 3.844 | 0.013 | 0.785 | StableMoFusion-10 | 3.862 | 0.013 | 0.781 |
> > | PMG-5 | 3.816 | 0.009 | 0.841 | PMG-10 | 3.893 | 0.011 | 0.830 |
> > | MoFA-5 | 3.947 | 0.011 | 0.852 | MoFA-10 | 3.961 | 0.014 | 0.820 |
> >
> > The table reports Foot Sliding (FS), Joint Smoothness (JS), and Motion Fluency (MF) for four models (MDM, StableMoFusion, PMG, MoFA) under two guidance settings (5 keyframes and 10 keyframes), $\textbf{without using contact loss. }$Across all metrics and both keyframe conditions, MDM achieves the best scores. However, this does not imply that MDM is inherently superior. Rather, MDM tends to generate more conservative motions with smaller amplitude, which naturally results in lower accumulated foot sliding and higher smoothness/fluency. When interpreted together with FID and MPJPE results in the appendix, MDM does not outperform other methods in overall motion realism or precision.
> >
> > The differences among the four models in FS, JS, and MF are relatively small, indicating that our MoFA will not produce noticeable or systematic foot-sliding artifacts. These metrics therefore help confirm that the proposed framework does not unintentionally bias the generation toward unstable contact or discontinuous motion.
> >
> > $\textbf{E2: What happened if we introduce foot-contact loss?}$
> > | Method | FS(cm) ↓ | JS ↓ | MF → | Method | FS(cm) ↓ | JS ↓ | MF → |
> > |--------|---------|------|------|--------|---------|------|------|
> > | **MDM-5** | **0.557** | **0.003** | **0.890** | MDM-10 | 0.565 | **0.003** | **0.895** |
> > | MoFA-5 | 0.558 | 0.004 | 0.781 | **MoFA-10** | **0.562** | 0.005 | 0.747 |
> >
> >
> > The table reports Foot Sliding (FS), Joint Smoothness (JS) and Motion Fluency (MF) for MDM and MoFA when contact loss is applied (using the same formulation as in MDM). We evaluate both models under 5-keyframe and 10-keyframe guidance.
> >
> > Compared with the table without contact loss, FS drops substantially for both models, indicating that contact loss is effective in reducing foot–ground drifting. Importantly, our model adapts to this constraint without architectural modification and remains close to MDM in all three metrics across both guidance settings. MDM still scores slightly better overall, which is consistent with the explanation provided earlier: MDM tends to generate more conservative, low-amplitude motions, naturally resulting in lower accumulated FS and higher smoothness/fluency.
> >
> > We also observe that adding contact loss generally improves JS (less abrupt acceleration) while slightly decreasing MF (reduced long-range temporal continuity). This trade-off is intuitive: adding a physical constraint stabilizes ground contact and suppresses sudden movements, but also mildly restricts the temporal variability of motion. Crucially, the differences remain small, and both models produce stable contact and continuous motion, demonstrating that the proposed method does not introduce foot-sliding artifacts and is compatible with contact-based physical regularization.

---

> > > ### Comment · Reviewer_7uQa · 2025-11-26
> > >
> > > Thank you for your hard work and sincere response. Much appreciated. Probably, this would be the last question.
> > >
> > > I now acknowledge that MoFA can be boosted by a foot contact loss, in terms of FS and JS scores. This is a good signal. The question I have now is how the foot contact loss affects the five metrics used in the main experiment: MPJPE, FID, Diversity, Traj.err, and K-MPJPE. Are they deteriorated? Or, is there no significant impact?

---

> > > > ### Author Response · Authors · 2025-11-26
> > > > **How the foot contact loss affects the five metrics.**
> > > >
> > > > Thank you for the constructive discussion. Here we report MPJPE, FID, Diversity, Traj.err, and K-MPJPE for MDM and MoFA under 5-keyframe and 10-keyframe guidance, ***without and with the contact loss (w/o C. and w/ C.)***. Consistent with previous observations, the 10-keyframe setting generally outperforms the 5-keyframe setting, and MoFA achieves stronger overall accuracy than MDM across both. This confirms that the results shown here are aligned with the rest of our evaluation.
> > > >
> > > > | Method | MPJPE ↓ | FID ↓ | Diversity → | Traj.err ↓ | K-MPJPE ↓ |
> > > > |--------|---------|--------|-------------|-------------|-----------|
> > > > | MDM-5 w/o C. | 4.582 ± .084 | 1.133 ± .045 | ***(4.378 ± .026)*** | ***(0.346 ± .018)*** | 4.418 ± .083 |
> > > > | **MDM-5 w/ C.** | **1.887 ± .039** | **0.328 ± .024** | 3.004 ± .037 | 0.457 ± .019 | **1.825 ± .036** |
> > > > | MoFA-5 w/o C. | 3.285 ± .096 | 0.459 ± .028 | ***(4.416 ± .028)*** | ***(0.064 ± .005)*** | 2.848 ± .091 |
> > > > | **MoFA-5 w/ C.** | **1.690 ± .040** | **0.262 ± .019** | 2.973 ± .034 | 0.163 ± .006 | **1.458 ± .028** |
> > > > | **MDM-10 w/o C.** | 4.136 ± .059 | 1.064 ± .043 | ***(4.413 ± .021)*** |***(0.329 ± .011)*** | 4.124 ± .057 |
> > > > | **MDM-10 w/ C.** | **1.738 ± .057** | **0.302 ± .022** | 3.011 ± .042 | 0.455 ± .016 | **1.733 ± .058** |
> > > > | MoFA-10 w/o C. | 2.204 ± .033 | 0.227 ± .011 | ***(4.431 ± .020)*** | ***(0.064 ± .003)*** | 1.901 ± .024 |
> > > > | **MoFA-10 w/ C.** | **1.152 ± .030** | **0.138 ± .010** | 2.938 ± .037 | 0.163 ± .007 | **1.050 ± .026** |
> > > >
> > > > Comparing **w/ contact loss vs. w/o contact loss for the same model and same guidance setting**, we observe substantial improvements in both **MPJPE** and **FID**, indicating that contact loss helps the model produce motions that are closer to the ground-truth distribution and reduces unrealistic details. At the same time, **Diversity** and **Traj.err** slightly decrease. This trade-off is reasonable: contact-based regularization reduces foot sliding and improves physical realism, which naturally narrows the range of lcoal motions and trajectories to lead to lower diversity, and makes the model less prone to "over-following" preplanned paths. Importantly, even with this reduction, the absolute values of Traj.err remains competitive and do not indicate a degradation of motion naturalness.
> > > >
> > > > Overall, these results show that (1) introducing contact loss consistently improves realism-oriented metrics (MPJPE, FID), (2) MoFA remains competitive under both settings without additional architectural changes, and (3) the changes in Diversity and Traj.err reflect a meaningful physical trade-off rather than a failure mode. This supports that our approach is compatible with contact-based constraints and benefits from them in a predictable and interpretable manner.

---

> > > > > ### Comment · Reviewer_7uQa · 2025-11-27
> > > > >
> > > > > Thank you for the additional experimental result.
> > > > >
> > > > > I am satisfied knowing that MoFA can be boosted by the foot contact loss in terms of FID, K-MPJPE, etc. This amount of degradation in Traj.err is reasonable for me, provided that motion quality improves.
> > > > >
> > > > > I am considering raising my rating. Would it be possible to include a discussion of the foot contact loss in the manuscript?

---

> > > > > > ### Author Response · Authors · 2025-11-27
> > > > > >
> > > > > > Thank you very much for considering raising the score and for your series of valuable comments. This exchange has been truly meaningful for us, making our arguments more rigorous and further improving the overall quality of the paper. Following your suggestions, we have added a detailed discussion of the foot-contact loss in the revised version, including additional clarification of the metrics, an explanation of the loss function, an experimental table, and an analysis of the results. All newly added content has been organized in Appendix C.6.

---

> > > > > > > ### Comment · Reviewer_7uQa · 2025-11-27
> > > > > > >
> > > > > > > Thank you for your reaction. I have raised my rating. Let me continue monitoring discussions with the other reviewers.
> > > > > > >
> > > > > > > One minor comment is that, as Reviewer 3qHX and I pointed out, some figures are too small. It would be good to fix it in the next revision.

---

> > > > > > > > ### Author Response · Authors · 2025-11-27
> > > > > > > >
> > > > > > > > Thank you very much for raising the score. We will consider how to further refine the pictures in future revisions. Likewise, we are also waiting for feedback from the other reviewers and look forward to hearing their suggestions as well.

---

> > > > > > > > ### Author Response · Authors · 2025-11-27
> > > > > > > >
> > > > > > > > We adjusted the layout of the oversmall image and added a tenth page to explore the relationship between the model's functionality and foot contact loss.

---

### Meta-Review · Area_Chair_ZQv6 · 2025-12-29

**Summary:**

**Summary**:
This paper presents MoFA, a diffusion-based motion factorization for controllable human motion synthesis.
The key innovation lies in two folds: combining the high-quality and low-quality data for the joint training and leveraging the local motion completion and global trajectory adaptation.
While the incremental improvement on the quantitative results is reported, the qualitative improvement is not obvious.
More importantly, a fair comparison on the same dataset is missed.

**Main strengths**:
- The proposed framework is well-designed for local motion completion and global trajectory adaptation.
- The motivation of utilizing both high-quality and low-quality data is interesting.

**Main weaknesses**:
- The unpaired control is not well demonstrated.
- The key novelty of local motion and global trajectory has been widely used in other works.
- A fair comparison is missed by using the same dataset.
- Many figures are too small to verify its key contribution.

**Suggested decision**: The paper received initial scores of 4 (7uQa), 6 (R7HS), 4 (UtVH), and 4 (3qHX), with reviewer 7uQa raising the score during the discussion. However, many concerns are not fully addressed, and the rebuttals are not well structured, making it very hard to understand the discussion and point. Hence, I recommend the final score as "reject".

**Reviewer Concerns:**

**Foot sliding is large (7uQa)**: Addressed.

**Some figures are quite small (7uQa, 3qHX)**: Still a big problem in the revised version.

**Evidence for "unpaired control" robustness is limited (R7HS)**: Additional results are provided, but the answer and conclusion are hard to buy from these results.

**Fair comparisons with the same training dataset (R7HS, UtVH)**: Additional results are provided in Appendix C.2.1, which should be included in the main paper. However, from Tables 12 and 13, the benefits of the proposed method are not obvious.

**Results to in-the-wild or longer sequences (R7HS)**: Does not support.

**Local motion and global trajectory alignment have been widely used in motion transfer (UtVH, 3qHX)**:  Not fully addressed.

**Description of test dataset (UtVH, 3qHX)**: Not fully addressed.

**The improvement over ProMoGen is not obvious (3qHX)**: "The goal of our work is not to outperform existing models by a large margin", which makes further confusion.

**Reviewer Scores:**

The paper initially received scores of 4 (7uQa), 6 (R7HS), 4 (UtVH), and 4 (3qHX).
During the discussion,
reviewer 7uQa raised the score,
while reviewer 3qHX claimed "your rebuttal has not adequately addressed my concerns."
As a result, I assume one reviewer would change the score in the dicsussion.

---

### Decision · Program_Chairs · 2026-01-26

Reject